# TestExplora: Benchmarking LLMs for Proactive Bug Discovery via Repository-Level Test Generation

**Steven Liu** [1 ‡]  **Jane Luo** [1 ‡]  **Xin Zhang** [1 †]  **Aofan Liu** [1 2 ‡]  **Hao Liu** [1 3 ‡]  **Jie Wu** [1 4 ‡]  **Ziyang Huang** [3]
**Yangyu Huang** [1]  **Yu Kang** [1]  **Scarlett Li** [1]

## Abstract

Given that Large Language Models (LLMs) are increasingly applied to automate software development, comprehensive software assurance spans three distinct goals: regression prevention, reactive reproduction, and proactive discovery. Current evaluations systematically overlook the third goal. Specifically, they either treat existing code as ground truth (a compliance trap) for regression prevention, or depend on post-failure artifacts (e.g., issue reports) for bug reproduction—so they rarely surface defects before failures. To bridge this gap, we present TestExplora, a benchmark designed to evaluate LLMs as proactive testers within full-scale, realistic repository environments. TestExplora contains 2,389 tasks from 482 repositories and hides all defect-related signals. Models must proactively find bugs by comparing implementations against documentation-derived intent, using documentation as the oracle. Furthermore, to keep evaluation sustainable and reduce leakage, we propose continuous, time-aware data collection. Our evaluation reveals a significant capability gap: state-of-the-art models achieve a maximum Fail-to-Pass ($F2P$) rate of only 16.06%. Further analysis indicates that navigating complex cross-module interactions and leveraging agentic exploration are critical to advancing LLMs toward autonomous software quality assurance. Consistent with this, SWEAgent instantiated with GPT-5-mini achieves an $F2P$ of 17.27% and an $F2P@5$ of 29.7%, highlighting the effectiveness and promise of agentic exploration in proactive bug discovery tasks.

## 1. Introduction

As Large Language Models (LLMs) are increasingly applied to automate software development (OpenAI, 2024; Gemini-Team, 2025; Qwen-Team, 2025), automatic test case generation has emerged as a pivotal area of research. In particular, comprehensive software assurance contains three distinct goals: regression prevention, reactive reproduction, and proactive discovery. Recent works have demonstrated that LLMs are effective at locking in existing behaviors to prevent regression, and at reproducing reported failures. However, their utility for proactively exploring latent bugs remains largely unexplored. Real-world software quality requires not only preserving the status quo or patching known defects but also identifying latent bugs that reside in repos.

As shown in Table 1, an examination of current benchmarks reveals that their design systematically overlooks this discovery capability by confining the evaluation to regression prevention and reactive reproduction. This limitation manifests in two primary ways. First, regarding regression prevention, a significant line of work constrains models to be reactive to implementation (Li & Yuan, 2024; Wang et al., 2025a; Zhang et al., 2024; Jain et al., 2025). By providing the source code as the ground truth and evaluating based on coverage, these benchmarks implicitly assume the correctness of the current implementation. The resulting tests act as characterization tests which codify existing behaviors to prevent future regressions. While useful for maintenance, it *potentially risks validating latent bugs as features.*

Second, even within the repository-level domain, where the necessary complexity exists, current benchmarks remain constrained by a reactive reproduction framing. Recent studies strictly leverage explicit issue reports (Hasan et al., 2025; Nashid et al., 2025) or specific commit changes (Pradel, 2025; Liu et al., 2025). Reflecting this focus, SWT-Bench (Mündler et al., 2024) and TDD-Bench-Verified (Ahmed et al., 2024) require a model to produce a unit test that replicates a pre-defined bug scenario. While effective for verification, this framing relies heavily on explicit issue reports or focused bug contexts, functioning primarily as post-hoc validation, which overlooks a critical reality: by the time defects are identified and documented, they have

---

[†]Corresponding author    [‡]Work done during internships at Microsoft  [1]Microsoft [2]Peking University [3]Chinese Academy of Sciences [4]Tsinghua University. Correspondence to: Xin Zhang <xinzhang3@microsoft.com>.

*Table 1.* Comparison of different open-source benchmarks with ours. Abbreviations: Cov = Coverage, Acc = Accuracy, MS = Mutation Score, ComR = Commit Relevance, FP = Fail-to-Pass Rate, CC = Change Coverage, Wb = White Box, Bb = Black Box.

| Benchmark | #Tasks | Task Level | Repos | Paradigm | Testing Scenario | Eval. Metrics |
|---|---|---|---|---|---|---|
| TestEval (Wang et al., 2024) | 216 | Function | — | Regression Prevention | Wb | Cov |
| UnL.Testbench (Huang et al., 2025) | 3909 | Function | — | Regression Prevention | Wb | Pass@k, MS |
| TestBench (Zhang et al., 2024) | 108 | Class | — | Regression Prevention | Wb | Cov, Acc, MS |
| TestGenEval (Jain et al., 2025) | 1210 | File | 11 | Regression Prevention | Wb | Pass@k, Cov, MS |
| ProjectTests (Wang et al., 2025b) | 295 | Project | 60 | Regression Prevention | Wb | ComR, Corr |
| SWTBench (Mündler et al., 2024) | 1900 | Project | 12 | Reactive Reproduction | Wb | FP, CC |
| TDDBench (Ahmed et al., 2024) | 449 | Project | 12 | Reactive Reproduction | Wb | FP, CC |
| **TestExplora (Ours)** | **2389** | **Project** | **482** | **Proactive Discovery** | **Bb & Wb** | **Cov, Acc, FP, CC** |

often already escaped into production, triggering costly incidents (e.g., data loss in large-scale cloud services) (Liu et al., 2019; Cotroneo et al., 2019; Ghosh et al., 2022; Jiang et al., 2024). *Relying on explicit issue reports is inherently reactive, as these reports are typically post-failure artifacts generated only after the damage has already occurred.* Additionally, most existing benchmarks rely on static datasets derived from SWE-Bench (Jimenez et al., 2024), *which cannot be updated timely and may introduce leakage.*

To address these limitations, we present TestExplora, a benchmark designed to evaluate LLMs as proactive testers within full-scale, realistic repository environments. First, to enable proactive discovery, TestExplora utilizes documentation as the reference oracle instead of specific defect reports. This forces models to identify latent bugs based on intended design specifications, breaking the dependency on prior knowledge of defects (Luo et al., 2025). Second, transcending the constraints of isolated code units, TestExplora operates at the repository level, challenging models to navigate complex dependencies and cross-module interactions to generate effective tests. Finally, addressing the risks of data leakage in static benchmarks, the proposed framework operates autonomously, enabling timely data updates from continuous repository streams. The evaluation harness of TestExplora executes generated tests on both buggy and fixed versions of the code. A test is considered valid if it fails before the fix and passes after, aligning with the Fail-to-Pass principle of defect validation. Beyond Fail-to-Pass rates, TestExplora also measures change-focused coverage. Our work has three contributions:

- **TestExplora benchmark:** We introduce TestExplora, the first benchmark shifting focus from regression prevention and reactive reproduction to largely unexplored proactive discovery. Unlike paradigms that rely on ground-truth code or explicit issue reports, TestExplora utilizes documentation-derived intent to challenge models to uncover latent defects across 2,389 tasks in 482 repos.

- **Comprehensive Evaluation:** Empirical results showing that even the strongest model achieves only a maximum Fail-to-Pass rate of 16.06%, posing a critical capability gap. Further analysis reveals that generating more tests does not necessarily lead to better performance. Addition-

ally, dependency preferences are highly model-specific, and agentic exploration is the key for proactive discovery.

- **Scalable benchmark collection framework:** To mitigate data leakage, we propose a pipeline that extracts and validates recent PRs in sanitized environments, yielding a time-aware stream for clean evaluation. And our code is available at GitHub[1]. And our data is at Hugging Face[2].

## 2. Related Work

**Benchmarks for Software Engineering.** Recent repo-level benchmarks focus on realistic, long-context, and multi-file dependencies. SWEBench (Jimenez et al., 2023) provides 2,294 issues and fixes from 12 Python projects and has become a widely used benchmark. Its extensions (Li et al., 2025; Chowdhury et al., 2024; Yang et al., 2024b; Zhang et al., 2025; Zan et al., 2025) further expand scale, difficulty, and dynamism to mitigate contamination and static overfitting. Other efforts include USEbench, which integrates multiple SWE tasks (Applis et al., 2025); DevEval, which evaluates multiple stages of the development lifecycle (Li et al., 2024); and other benchmarks (Le Hai et al., 2024; Zhao et al., 2025), which target completion, dependency handling, and real-world bug fixing.

**Benchmarks for Test Generation.** Driven by the growing importance of testing, subsequent work has explored leveraging LLMs for test generation (Wang et al., 2025d; Hasan et al., 2025; Wang et al., 2025c). At the function level, TestEval and UnL.Testbench (Wang et al., 2024; Huang et al., 2025) focus on coverage while addressing contamination and realism. At the class level, TestBench (Zhang et al., 2024) samples 108 Java classes for evaluation. At the project level, CLOVER (Xu et al., 2025) examines long-context generation, ProjectTest (Wang et al., 2025b) targets medium-scale projects across three languages, and TestGenEval (Jain et al., 2025) builds on 68k human-written tests to assess writing, completion, and improvement. SWT-Bench (Mündler et al., 2024) and TDD-Bench Verified (Ahmed et al., 2024) tie tests to real issues and fixes.

---

[1] https://github.com/microsoft/TestExplora
[2] https://huggingface.co/datasets/microsoft/TestExplora

**Classical Test Generation.** Prior to LLM-based approaches, automated test generation has been extensively studied in the software engineering community. Search-based methods such as EvoSuite (Fraser & Arcuri, 2011) and Pynguin (Lukasczyk & Fraser, 2022) generate test suites by optimizing coverage and mutation-based fitness functions. Property-based testing frameworks such as QuickCheck (Claessen & Hughes, 2000) shift the oracle from concrete assertions to user-specified invariants. These approaches typically rely on the existing implementation as the oracle and thus are not designed to surface latent defects that violate intended behavior.

## 3. TestExplora

TestExplora is a benchmark designed to measure large language models' ability in exploratory software testing—specifically, their capacity to proactively discover defects rather than merely reproduce known errors. The subsequent sections present across three components: Benchmark Acquisition, Model Inputs, and Evaluation Metrics.

### 3.1. Benchmark Acquisition

All instances of TestExplora are derived from existing GitHub repositories to ensure data quality. During the construction of TestExplora, carefully maintained repositories were selected. As illustrated in Figure 1, the Benchmark Acquisition process primarily comprises the following steps:

**Step1.Crawling Raw Issue-PR** Constructing a reliable exploratory testing benchmark requires high-quality, well-maintained codebases where defects can be meaningfully discovered and validated. We select only repositories $\mathcal{R}$ with substantial community adoption (>1,000 stars), yielding 12,227 preliminary pull requests with patches $\mathcal{P}$ that provide genuine opportunities for proactive defect detection.

**Step2.Filtering Validated PR** For each patch $\mathcal{P}$, we can decompose it into a test patch $\mathcal{P}_t$ and a code patch $\mathcal{P}_c$, where $\mathcal{P}_c = \mathcal{P}/\mathcal{P}_t$. By employing the parser $parser(\cdot)$, we can identify the function with the number of $n$ modifications involved in each patch, such that $parser(\mathcal{P}_k) = \{f_{1,k}, \ldots, f_{n,k}\}, k \in \{c, t\}$. Before applying the Fail-to-Pass filtering, we first analyze the invoke graph of the repository. We retain only PRs where **repaired functions $f_c \in parser(\mathcal{P}_c)$** are reachable by **test functions $f_t \in parser(\mathcal{P}_t)$** through explicit call paths. This aims that our benchmark captures realistic testing scenarios where generated tests can meaningfully exercise the modified code and potentially discover defects through proper invocation chains. This step differs from previous benchmarks (Mündler et al., 2024; Ahmed et al., 2024), which only applied execution-based filtering without verifying whether the assertion was indeed caused by the code patch $\mathcal{P}_c$. In

addition, we filter out PRs in which $\mathcal{P}_c$ involves doctoring modifications, in order to ensure that the function's intention does not undergo substantial changes before and after the pr. We also exclude PRs where, after preprocessing, $parser(\mathcal{P}_c)$ contains doctoring with "TODO" annotations or implementations including "pass," thereby ensuring that the code after the pr does not contain potential placeholders for future updates. Finally, we designate the functions directly invoked by $parser(\mathcal{P}_t)$ as **entry interface $\mathcal{E}$**, which are then used for subsequent test generation across pipelines.

**Step3. Automated Build Virtual Machine** Constructing Docker environments for large-scale repositories is challenging; therefore, we adopted GitHub Actions Runner Images[3] to establish a unified virtual machine with test scripts, enabling Fail-to-Pass testing in a manner analogous to GitHub Actions Runners. Action Run script is listed in the Appendix A. During the setup of each repository for testing, the process mainly consists of the following three steps: **a) Install system dependencies**: This step primarily prepares the system-level compilation toolchain and external library dependencies. **b) Look for and merge all requirements files**: It automatically detects and merges requirements within the project, ensuring that no requirements are missed or duplicated during subsequent installation. **c) Environment Setup**: This step selects the logic for dependency installation based on the project structure in order to configure the testing environment. The script attempts to identify a pyproject.toml project and installs dependencies using Poetry or PDM. If the project is not a pyproject project, it installs dependencies using pip together with requirements/setup.py and common utility packages. *This approach facilitates flexible setup of the testing environment, and such flexibility enables us to expand existing datasets at any time without manual setup enabling the extensibility of our framework.*

**Step4. Reference Oracles** We adopt an execution-based approach to ensure that the bug is indeed detected by the generated test. Specifically, our framework conducts golden tests on $\mathcal{P}_t$ both before and after applying the code patch $\mathcal{P}_c$. Ultimately, our dataset comprises 1,552 pull requests and 2,389 test generation tasks across 482 repositories. The Appendix G provides information on the categories of the repositories and related details.

When generating tests for $\mathcal{E}$, it is essential to clarify the intention of $\mathcal{E}$ to enable effective exploratory testing that can uncover potential defects. The most straightforward approach is to leverage existing docstrings of $\mathcal{E}$ as documentation, enabling LLMs to infer function intent and generate comprehensive tests. However, we observe that adequate documentation is rarely available for entry interface in real-world repositories. To address this documentation scarcity, we use a high-performing agent—DocAgent (Yang et al.,

---

[3]https://github.com/actions/runner-images

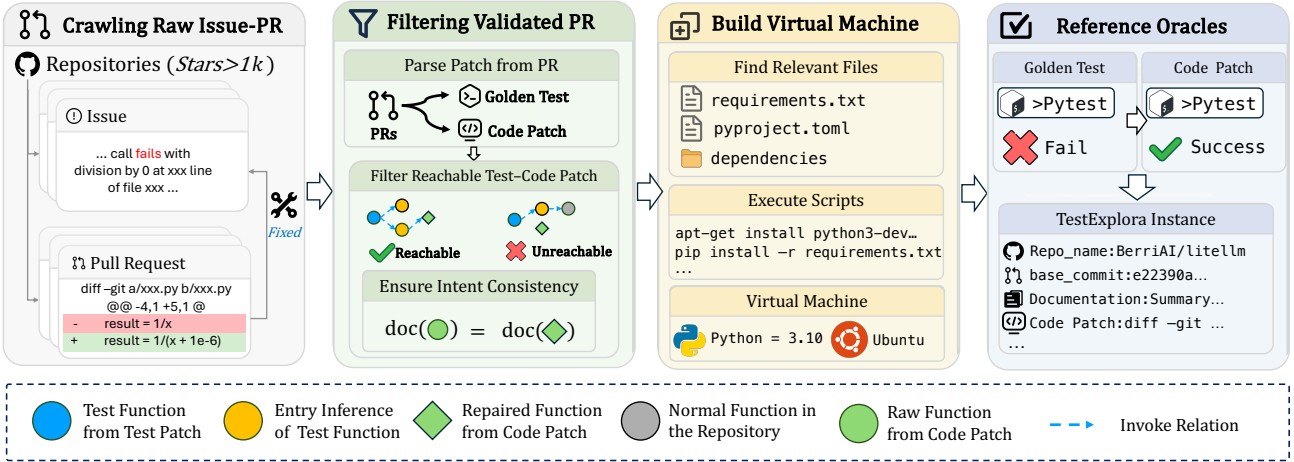

*Figure 1.* The data acquisition process of TestExplora. Different from SWTBench (Mündler et al., 2024), TestExplora is characterized by its reliance on documentation rather than issue descriptions and requires models to proactively identify bugs that violate the documentation, instead of reactively reproducing known failures. Moreover, unlike SWTBench, which is derived from SWEBench (Yang et al., 2024b), *the repositories we collect are mutually exclusive from SWEBench and are designed to be extensible.*

*Table 2.* The statistical information of TestExplora. **Categories** denotes the numbers of repository categories of repositories. In **Test Invokes**, **Entries per Test** counts functions invoked by a test case, while $\mathcal{P}_c$ **Depth** is the invocation distance between the test case and the modified code patch.

| Corpus Overview | | | | |
|---|---|---|---|---|
| **Repositories** | **PRs** | **Tests** | **Categories** | **Avg. Stars** |
| 482 | 1552 | 2389 | 21 | 5010.77 |

| PR Instance Statistics | | | | |
|---|---|---|---|---|
| **Area** | **Indicator** | **Max** | **Mean** | **Median** |
| Test Patch | # Tests Edited | 23 | 1.60 | 1 |
| Test Invokes | Entries per Test | 154 | 8.22 | 3 |
| | $\mathcal{P}_c$ Depth | 12 | 1.76 | 1 |
| Entry Interface | # Dependencies | 150 | 3.57 | 1 |
| | Lines of Code | 1378 | 29.07 | 13 |
| Code Patch | # Lines Edited | 1297 | 22.92 | 11 |
| | # Functions Edited | 29 | 1.42 | 1 |
| | # Files Edited | 28 | 1.83 | 1 |

2025)—to generate the corresponding documentation for $\mathcal{E}$. This automated annotation approach greatly enhances the scalability of TestExplora. We list a log from the execution process of DocAgent in the Appendix B. Following the above steps, TestExplora consists of 1,552 pull requests from 482 repositories, along with 2,389 test cases as generation tasks. Table 2 presents the information of TestExplora.

## 3.2. Model Inputs

Effective exploratory test generation requires strategic information provisioning to balance realism with model capability. TestExplora provides three types of input information:

- **Documentation**: Documentation, produced through the two-stage generation process described in Section 3.1,

is employed to clarify what the test entry inferences are intended to accomplish, what their inputs and outputs are, and what potential errors may arise.

- **Test Entry Inferences**: As mentioned in Section 3.1, test entry inferences $\mathcal{E}$ are codes that are directly invoked by the test cases $parser(\mathcal{P}_t)$.

- **Dependencies**: Dependencies are the direct dependencies of the test entry inferences and are used to further clarify the intention of the test entry inferences for the LLMs.

To better simulate testing under different conditions, we defined two testing scenarios: White Box and Black Box. As illustrated in the lower-right corner of Figure 2, these scenarios differ in terms of the input information provided. Specifically, Code Imp. indicates whether the concrete implementation of the test entry inferences is provided, while Dep. denotes whether the dependencies of the test entry inferences are included. In all two settings, the documentation (Doc.) of the test entry inferences is consistently provided.

## 3.3. Evaluation Metrics

Before introducing the evaluation metrics, we first formalize the test generation task: $\mathcal{T}_{n,t}^* = \Theta(\mathcal{E}_{n,t}|s), s \in \{\text{White Box, Black Box}\}$ are tests generated by the model $\Theta(\cdot|\cdot)$ given the test entry inferences $\mathcal{E}_{n,t}$. $\mathcal{E}_{n,t}$ denotes the entry inferences of a test $t$ in the ground truth tests $\mathcal{T} = parser(\mathcal{P}_t)$ from the $n^{th}$ data snippet's test patch $\mathcal{P}_{n,t}$. To evaluate the quality of $\mathcal{T}_n^* = \bigcup_{t \in \mathcal{T}} \mathcal{T}_{n,t}^*$, we employed the following metrics:

## 3.4. Test-based Metrics

The motivation behind test-based metrics is: 1)the generated test cases must align with the intended behavior ($HP$), and

*Table 3.* Performance comparison of different models on TestExplora. The best results are in **bold**, and the second-best are underlined.

| | | TestExplora | | | | | TestExplora-Lite | | | | |
|---|---|---|---|---|---|---|---|---|---|---|---|
| Type | Model | $HP$ | $F2P$ | $EC$ | $CFG$ | $Num.$ | $HP$ | $F2P$ | $EC$ | $CFG$ | $Num.$ |
| Black Box | Qwen3-Coder-30B | 63.98 | 2.05 | **52.98** | 42.21 | 19.48 | 54.38 | 2.30 | 49.98 | 42.21 | 13.40 |
| | o3-mini | 68.27 | 3.93 | 51.46 | 41.67 | 13.86 | 58.37 | 5.58 | 56.43 | 44.64 | 8.39 |
| | o4-mini | 66.18 | 2.59 | 51.18 | 43.24 | 14.19 | 56.97 | 3.74 | 59.44 | **46.29** | 10.82 |
| | Gemini-2.5-pro | 62.04 | 2.06 | 38.40 | 41.45 | 14.26 | 58.74 | 2.60 | 53.24 | 44.91 | 12.89 |
| | GPT-4o | 62.58 | 1.47 | 46.98 | 42.14 | 12.31 | 51.38 | 3.61 | 54.69 | 43.55 | 7.75 |
| | GPT-5-mini | **73.28** | **5.17** | 52.42 | **43.41** | 11.50 | **60.99** | **7.67** | **59.72** | 45.59 | 7.96 |
| White Box | Qwen3-Coder-30B | 68.80 | 1.39 | 53.99 | 43.01 | 19.21 | 60.90 | 2.09 | 59.78 | 44.05 | 13.21 |
| | o3-mini | 76.54 | 4.19 | 52.49 | 42.16 | 12.68 | 71.53 | 7.07 | 61.84 | 43.78 | 8.00 |
| | o4-mini | 72.33 | 5.92 | **67.30** | **47.06** | 10.48 | 74.70 | 6.45 | **67.16** | **46.32** | 10.36 |
| | Gemini-2.5-pro | 74.51 | 2.59 | 45.88 | 43.05 | 16.23 | 74.91 | 5.08 | 61.47 | 44.25 | 12.84 |
| | GPT-4o | 67.91 | 1.79 | 46.00 | 41.64 | 11.22 | 57.92 | 2.20 | 56.29 | 44.15 | 7.42 |
| | GPT-5-mini | **84.80** | **7.54** | 56.33 | 43.17 | 11.55 | **80.25** | **11.84** | 65.78 | 44.37 | 7.78 |

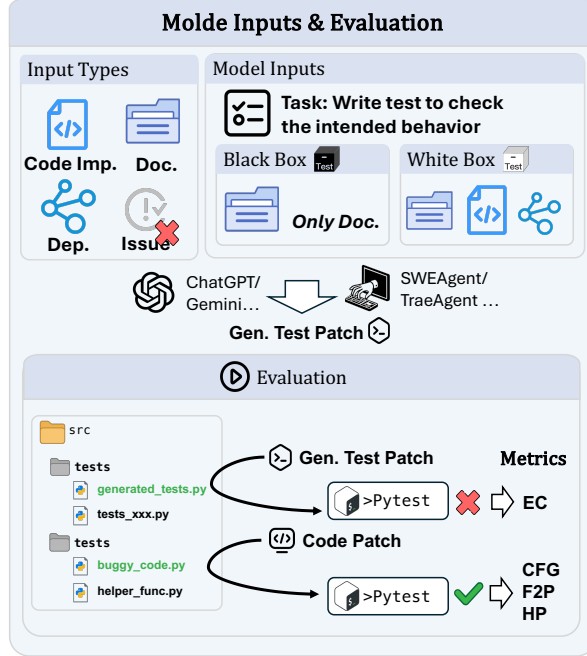

*Figure 2.* According to the differences in input information, the tests are mainly divided into two scenarios: White Box testing and Black Box testing, while performance is evaluated via four metrics: $HP$, $F2P$, $EC$ and $CFG$.

2) the generated test suite must expose bugs ($F2P$).

**Head Pass Rate (HP)**: We measured the pass rate of the tests generated by the model on the head commit after the pull request, *which reflects the accuracy with which the generated tests adhere to the intended behavior*. The Head Pass Rate is a test-level metrics:

$$HP = \frac{|\bigcup_{n=1}^{N}\{t \in \mathcal{T}_n^*|pass(t, \mathcal{R}_{n,base} \times \mathcal{P}_{n,c})|\}}{\sum_{n=1}^{N}|\mathcal{T}_n^*|}, \quad (1)$$

where $pass(\cdot, \cdot)$ serves as an indicative function $\mathbb{I}$, which

denotes whether the test $t$ passes on the head repository $\mathcal{R}_{n,base} \times \mathcal{P}_{n,c}$, where the issue from the base commit repository $\mathcal{R}_{n,base}$ is fixed with $\mathcal{P}_{n,c}$.

**Fail-to-Pass Rate (F2P)**: Similar to SWT-bench (Mündler et al., 2024), we compute the proportion of PRs for which at least one generated test exhibits a Fail-to-Pass transition. *This metric reflects the effectiveness of the generated tests, indicating the extent to which the model can accurately identify potential errors*:

$$F2P = \frac{\sum_{n=1}^{N} \mathbb{I}(|f2p(n)| \geq 1)}{N}$$
$$f2p(n) = \{t \in \mathcal{T}_{pr}^* | pass(t, \mathcal{R}_{n,base} \times \mathcal{P}_{n,c}) \quad (2)$$
$$\& fail(t, \mathcal{R}_{n,base})\}\},$$

where $f2p(\cdot)$ is a pr level function, which finds the test that passes on the head repo and fails on the base repo.

### 3.5. Code-based Metrics

The motivation behind code-based metrics is: 1) achieving comprehensive coverage of the code, and 2) ensuring coverage that is specifically aligned with recent code changes.

**Entry Coverage (EC)**: Entry Coverage measures the line coverage of the generated tests with respect to the test entry inferences, reflecting the extent to which the generated tests comprehensively capture the intention of the test entry inferences:

$$EC = \frac{1}{N} \sum_{n=1}^{N} \frac{|cover(\mathcal{T}_n^*, \mathcal{E}_n)|}{|line(\mathcal{E}_n)|}, \quad (3)$$

where $cover(\cdot, \cdot)$ denotes the set of lines from entry interface $\mathcal{E}_n$ covered by the tests $\mathcal{T}_n^*$. And $line(\cdot)$ returns the lines of $\mathcal{E}_n$.

**Change-focused Coverage (CFG)**: When applying $\mathcal{P}_{n,c}$, certain lines in $\mathcal{R}_n$ are modified, denoted as $\Delta(\mathcal{R}_n)$:

$$CFG = \frac{1}{N} \sum_{n=1}^{N} \frac{|cover(\mathcal{T}_n^*, parser(\mathcal{P}_{c,n}))|}{|\Delta(\mathcal{R}_n)|}. \quad (4)$$

The aforementioned metrics collectively capture the model's fidelity to the code's intended functionality, its precision in identifying bugs, and the effectiveness of the generated tests.

# 4. Evaluation

In the following sections, we want to answer the following research questions (**RQs**):

- **RQ1:** How do existing models perform on TestExplora?
- **RQ2:** What factors can influence the performance?
- **RQ3:** How can model performance be further improved?

We conduct a comprehensive evaluation utilizing four key metrics: Head Pass Rate, Fail-to-Pass Rate, Entry Coverage, and Change-focused Coverage. Additionally, the number of testcases ($Num.$) is reported. Our experiments involve six representative language models evaluated on 12,227 real-world pull requests. To ensure evaluation efficiency, we constructed a high-quality subset, *TestExplora-Lite*, by filtering samples based on the quality of human-written docstrings. This subset comprises 330 PRs and 517 samples in total. All baseline experiments are repeated three times to ensure the robustness of the results.

**Models** To comprehensively evaluate the problem detection capability of existing LLMs, we select six mainstream models. Among open-source code models, we select the Qwen3-Coder-30B-A3B (Qwen-Team, 2025). For general LLMs, GPT-4o (OpenAI, 2024) and GPT-5-mini[4] are selected. For reasoning models, TestExplora evaluates o3-mini, o4-mini[5], and Gemini-2.5-pro (Gemini-Team, 2025).

**Context Template** To comprehensively evaluate the capability of LLMs in problem detection, we adopt different input formats to simulate various testing scenarios. Specifically, White Box and Black Box are selected as two main testing scenarios. In the White Box scenario, models are allowed to access the complete codes along with their related dependencies. In contrast, in the Black Box scenario, the model is only provided with the corresponding test entry inferences and the associated documentation. The detailed templates are provided in Figure 2 and Appendix C.

---

[4]https://openai.com/index/introducing-gpt-5/
[5]https://openai.com/index/o3-o4-mini-system-card/

## 4.1. Evaluation Results

**Performance comparison among the models** As detailed in Table 3 regarding *RQ1*, GPT-5-mini achieves state-of-the-art performance, securing the best $F2P$ scores of 7.67 (Black Box) and 11.84 (White Box). Notably, it attains superior $EC$ and $CFG$ scores while generating fewer tests, underscoring its efficiency in capturing interface branches and locating errors. o4-mini follows with the second-best performance across these metrics. We observe that Gemini-2.5-pro tends to generate significantly more test cases than other models. Notably, while Gemini-2.5-pro achieves a relatively high $HP$ score, its $F2P$ remains comparatively low. Since $HP$ is an evaluation metric at the test-case level, this high $HP$ indicates that Gemini-2.5-pro has a strong fundamental ability to write individual correct tests. However, $F2P$ is measured at the test-suite level, where a test suite passes only if all its constituent test cases are correct. Generating more test cases per suite inherently increases the probability of at least one test being incorrect. For instance, if GPT-5-mini generates approximately 8 test cases per suite while Gemini-2.5-pro generates around 13, the latter must maintain correctness across a larger number of tests to achieve the same $F2P$. Consequently, despite its higher test-case-level quality, Gemini-2.5-pro's larger test volume dilutes its suite-level success rate. Overall, existing LLMs perform poorly on TestExplora, indicating that they lack the capability to proactively discover bugs.

Having established the varying efficacy of current models, we next investigate the underlying determinants driving these results (*RQ2*) by conducting a series of experiments from output and input perspective:

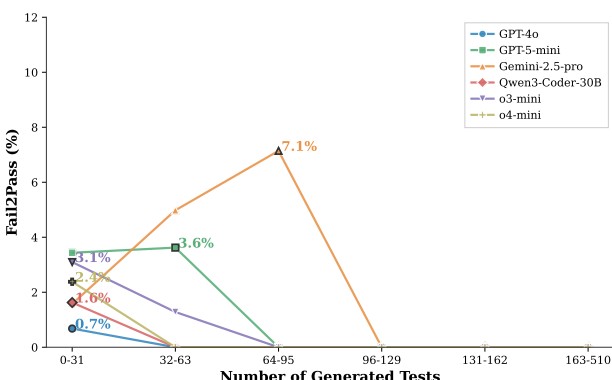

*Figure 3.* The impact of the number of generated test cases on performance. The best performance of each model is highlighted.

**Generating More Tests Does Not Necessarily Lead to Better Performanc** Previous results show that while Gemini-2.5-pro generates high-quality test cases, its tendency to produce a large number of tests negatively impacts the test-suite-level $F2P$. Figure 3 confirms this trend, showing that $F2P$ roughly decreases as the generated tests increases.

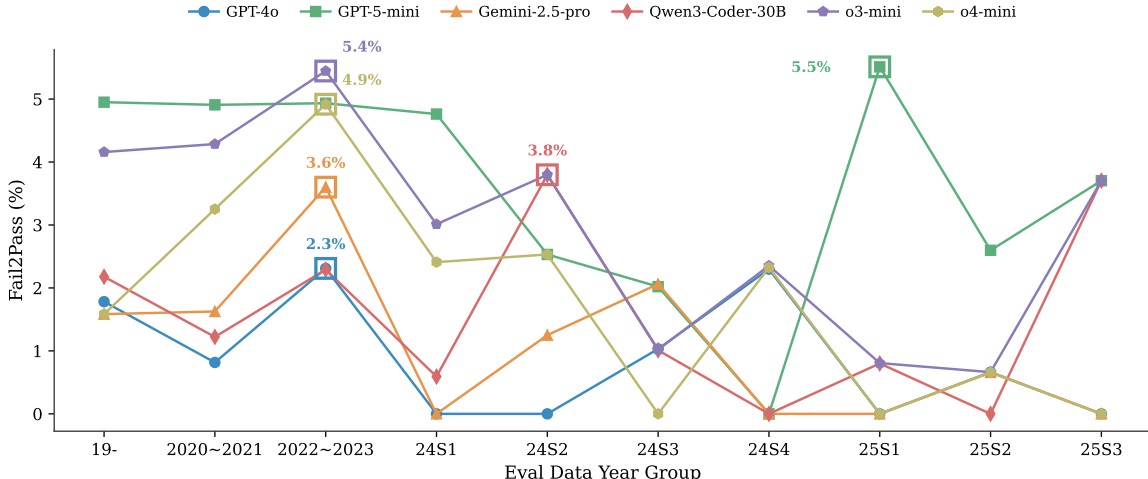

*Figure 4.* Fail-to-Pass success rates across instance year buckets for six LLMs without dependency code access, highlighting each model's peak performance season.

When the number of generated tests exceeds 96, no generated test suit could pass. The experimental results indicate that **generating more tests does not necessarily lead to better performance**. However, generating too few tests also poses potential risks of hacking. For example, in Table 3, GPT-4o produces fewer tests, but its coverage metric is comparatively lower, indicating that it may not generate sufficiently comprehensive tests. These results suggest that, at the repository level, **a test suite depends not only on a model's ability to generate tests, but also on its ability to orchestrate a concise yet high-coverage test suite**.

Beyond the quantity of generated output tests, the macroscopic distribution of input data also plays a critical role.

**Robustness Across Timelines via Scalability Framework** Figure 4 shows how the performance of models in test generation varies over time. We find that existing models perform better on data prior to 2023 than on data after 2023. Since SWEBench (Jimenez et al., 2024) covers pull requests from before 2023, existing models have been trained on repositories related to SWEBench, which leads to their relatively stronger performance on pre-2023 data. This indicates that existing datasets (Mündler et al., 2024; Ahmed et al., 2024) based on SWEBench may suffer from data leakage risks, while also **highlighting the importance of our scalability framework**. For GPT-4o, its performance remains relatively poor, which is because the selected model (2024-05-13) is not specifically trained on SWE-type datasets.

**Performance disparities exist across repository domains** We also analyze the impact of repository type on model performance. As shown in Figure 7, the models perform best in the Scientific/Engineering domain, where all models achieve an F2P score above 5%. Notably, both o4-mini and Qwen3-Coder fail the Security domain with scores of 0. In

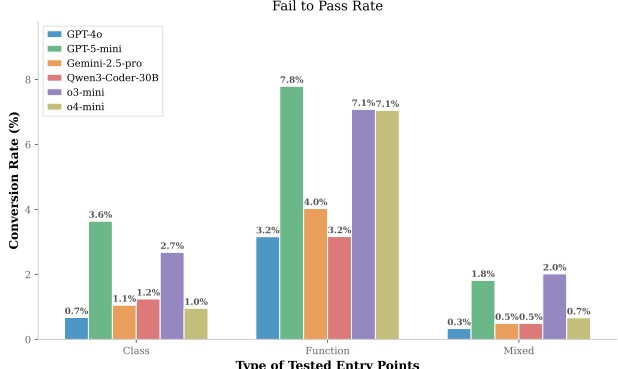

*Figure 5.* Fail-to-Pass success rates across test entry inferences type. Class indicates that the test entry inferences consist only of classes, Function indicates that they consist only of functions, and Mixed denotes a combination of both.

addition, the microscopic structural complexity of inputs at also affects the model's test generation performance.

**Performance variation with respect to the types of entry inferences** In Figure 5, all models achieve their peak performance on function-type code, which represents the simplest structural unit. Performance declines for class-based entry inferences and drops significantly for mixed involving both functions and classes. This stratification naturally segments TestExplora into three distinct difficulty levels: Easy (Function), Medium (Class), and Hard (Mixed).

**Optimal dependency context of entry inferences is highly model-dependent** In Table 3, we observe that for certain models—such as GPT-4o—their White Box performance in the Lite version is inferior to their Black Box performance. To investigate this issue, we hypothesize that GPT-4o may not effectively leverage contextual information. Therefore, in Figure 6, we compute the performance variations of dif-

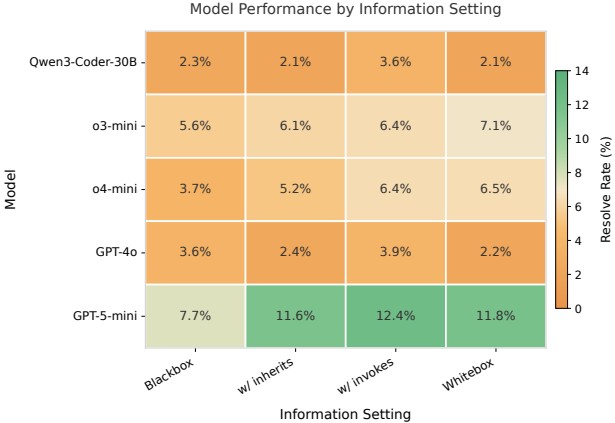

*Figure 6.* The performance of the model varies with the distribution of dependencies. Whitebox only inherits indicates that only inheritance dependencies are retained. Whitebox only invokes indicates that only invocation dependencies are retained.

ferent models under different types of dependencies. In our setting, we categorize dependencies of test entry inferences into two types: invocation and inheritance. Our analysis yields two key insights. First, extensive context is not universally beneficial: GPT-4o's F2P rate drops from 3.6% (Black Box) to 2.2% (White Box), suggesting a struggle to filter relevant details from complete dependency information. Second, models exhibit distinct dependency preferences. While GPT-4o favors invocation dependencies (3.9%) over inheritance (2.4%), o3-mini and o4-mini consistently benefit from full context, peaking in the White Box setting (7.1% and 6.5%, respectively). GPT-5-mini demonstrates the most robustness, reaching 12.4% with invocation dependencies.

*Table 4.* Performance comparison with different context methods in White Box context. The best results of each context are in **bold**.

| Context | Model | Model | | | | |
|---|---|---|---|---|---|---|
| | | $HP$ | $F2P$ | $EC$ | $CFG$ | $Num.$ |
| Plain | DeepSeek-R1 | 63.72 | 3.03 | 47.16 | 34.77 | 10.22 |
| | o4-mini | 72.33 | 5.92 | 67.30 | 47.06 | 10.48 |
| | GPT-5-mini | 77.68 | 12.79 | 67.76 | 45.66 | 7.69 |
| | Claude-Sonnet-4.5 | **87.52** | **16.06** | 71.96 | **47.37** | 23.94 |
| | GPT-5 | 84.88 | 15.15 | 69.44 | 45.59 | 9.59 |
| SWEAgent | o4-mini | 82.03 | 9.42 | **69.63** | **47.81** | 13.79 |
| | GPT-5-mini | **93.81** | **17.27** | 65.45 | 47.43 | 23.09 |
| TraeAgent | o4-mini | 86.55 | 12.16 | 62.01 | **47.95** | 12.76 |
| | GPT-5-mini | **88.50** | **14.55** | 62.01 | 46.84 | 12.79 |

**Agents as a solution of proactive testing:** From the previous section, our experimental results indicate that different models exhibit preferences for different types of dependencies. Therefore, we introduce the Agent baseline to answer *RQ3*. Specifically, we allow the SWEAgent (Yang et al., 2024a) and Trae-Agent (Team et al., 2025) to freely explore the repository to generate tests that uncover potential issues. From Table 4, we can observe that, compared to providing all dependencies directly, the Agent makes more efficient

use of the context. This suggests that, for the tasks represented by TestExplora, the Agent constitutes a promising research direction. We also analyze the tool-invocation frequency of different agents. We find that for SWEAgents based on GPT-5-mini and o4-mini, the tool used most frequently at every stage is str_replace_editor view. Figure 8 and Figure 9 indicate that the agents are consistently exploring the repository throughout the process. Moreover, the top 15 most frequently used tools are also predominantly focused on repository exploration and comprehension. In addition, we also compute the F2P pass@k metrics for SWEAgent in Figure 10, which shows that as the number of samples increases from 1 to 5, SWEAgent's $H2P$ rises to 29.7 with GPT-5-mini and to 21.2 with o4-mini, indicating that current agents still have substantial room for improvement. **The above results indicate that freely exploring the repository, rather than relying on simple input–output interactions, is more effective.**

## 5. Further Analysis

**Ablation on documentation** In TestExplora, we adopt DocAgent (Yang et al., 2025) as an additional source of information for test generation. To verify the effectiveness, we compare the performance differences between synthetic and human-written documentation in TestExplora-Lite.

*Table 5.* Performance change from human-written documentation to DocAgent-generated documentation. The table reports the information gain of DocAgent-generated documentation compared with human-written documentation.

| Type | Model | TestExplora-Lite Change | | | | |
|---|---|---|---|---|---|---|
| | | $HP$ | $F2P$ | $EC$ | $CFG$ | $Num.$ |
| Black Box | QC-30B-A3B | +13.42 | +1.07 | +0.39 | +0.19 | -0.16 |
| | o3-mini | +16.78 | +3.33 | +6.20 | +0.92 | +0.01 |
| | o4-mini | +6.55 | +1.82 | +3.28 | +0.17 | +0.27 |
| | Gemini-2.5-pro | +10.7 | +0.61 | +8.53 | +0.81 | +0.15 |
| | GPT-4o | +4.39 | +2.12 | +6.54 | +0.33 | +0.70 |
| | GPT-5-mini | +10.12 | +6.06 | +10.52 | +1.83 | +0.13 |

From Table 5, it can be observed that the documentation generated by DocAgent provides stronger informational gains compared to human-written documentation. Specifically, the documentation generated by DocAgent does not lead to significant changes in the average number of tests generated by the models. Instead, it helps improve performance across other metrics. Since DocAgent observes only the information from the head repository during exploration—without accessing diff patches or error messages—there is no risk of potential information leakage. The ablation study further demonstrates the scalability of TestExplora. To ensure effectiveness of documentation, we performed a rigorous human evaluation on 200 randomly sampled cases, confirming that the generated documentation accurately reflects intended functionality with no patch leakage. *We further conducted GPT-5 audit over 1,552 PRs, finding only 16 cases (1.03%)*

*of Patch Leakage and 6 (0.38%) of Task Supervision issues.*

**Error Analysis** We also conduct an analysis of the errors produced by models. From Figure 11, we observe that all models are more prone to *Assertion Mismatch errors and Misconfigured Mocks*. In other words, existing models fail to accurately capture the behavior and output of the function under test, both in Black Box and White Box scenarios. We also conduct case study in Appendix F.

## 6. Conclusion

We present TestExplora, an extensible new benchmark for evaluating LLMs in proactive software testing. TestExplora challenges models to proactively uncover defects in realistic repository-level contexts from documentation. Our evaluation reveals a critical capability gap: even the strongest models achieve less than 16% Fail-to-Pass success. Analysis shows model-specific dependency preferences and the superiority of agentic exploration over static context.

## Impact Statement

This paper presents work whose goal is to advance the field of Machine Learning. There are many potential societal consequences of our work, none of which we feel must be specifically highlighted here.

## Limitations

TestExplora uses DocAgent-generated documentation from the post-fix repository as its reference oracle. This isolates the capability we measure. The paper aims at detecting intent-implementation mismatches, separating this from the harder problem of recovering intent, which is rarely so cleanly available in practice. Reported F2P rates should thus be read as an upper bound on this sub-task.

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

# A. Action Run Script

This Action Run workflow automates the testing pipeline for Python projects. Upon each code push, it performs the following steps: 1) checks out the repository and verifies the Python environment; 2) installs necessary system-level dependencies; 3) automatically detects and merges all requirements files, removing duplicates; 4) **intelligently identifies the project's dependency management tool—supporting Poetry, PDM, PEP 517-compliant pyproject.toml, and traditional setup.py configurations—and installs dependencies accordingly**; and 5) executes the test suite using pytest while generating a JSON-formatted coverage report via the coverage tool. This workflow provides a unified, adaptive CI solution capable of handling diverse Python project structures without manual configuration:

**Action Run Script**

```
name: Tests
on:
  push

jobs:
  tests:
    runs-on: ubuntu-latest

    steps:
    - name: checkout
      uses: actions/checkout@v4

    - name: Detect Python
      run: |
        which python3
        python3 --version

    - name: Install system dependencies
      run: |
        sudo apt-get update
        sudo apt-get install -y python3-dev librados2 librados-dev libpq-dev build-essential

    - name: Look for and merge all requirements files
      id: find_reqs
      shell: bash
      run: |
        echo "===> Looking for requirements*.txt or *requirements.txt files..."
        find . -maxdepth 1 -type f \( -iname "requirements*.txt" -o -iname "*requirements.txt" \) > all_package.
txt

        if [[ -s all_package.txt ]]; then
          echo "Found requirements files:"
          cat all_package.txt
          echo "HAS_REQUIREMENTS=true" >> $GITHUB_ENV

          echo "===> Merging and deduplicating requirements..."
          touch merged-requirements.txt
          while read -r file; do
            grep -vE '^\s*#' "$file" | grep -vE '^\s*$' >> merged-requirements.txt || true
          done < all_package.txt

          sort merged-requirements.txt | uniq > requirements-all.txt
          echo "===> Combined requirements:"
          cat requirements-all.txt
        else
          echo "No requirements files found."
          echo "HAS_REQUIREMENTS=false" >> $GITHUB_ENV
        fi

        python3 -m pip install --upgrade pip

    - name: Install Python via pyproject
      run: |
        if [ -f "pyproject.toml" ]; then
          echo "===> pyproject.toml detected"
          echo "USE_POETRY=true" >> $GITHUB_ENV

          echo "===> install poetry"
          pip install poetry

          if [ -f "poetry.lock" ]; then
            echo "===> Checked poetry.lock and Using Poetry"
            poetry install || echo "Poetry install dev failed or timed out"
```

```
                elif [ -f "pdm.lock" ]; then
                    echo "===> Checked pdm.lock and Using PDM"
                    echo "USE_PDM=true" >> $GITHUB_ENV
                    pip install pdm
                    pdm install --with test || echo "Test group not found or failed"
                    pdm install --with dev || echo "Dev group not found or failed"
                else
                    if grep -q 'ˆ\[tool\.poetry\]' pyproject.toml || grep -q 'ˆ\[project\]' pyproject.toml; then
                        echo "===> This pyproject.toml is a project"
                        echo "===> No lock file detected, installing via pip (PEP 517)..."
                        echo "===> install poetry pytest-json-report"
                        poetry install --no-interaction --with dev || echo "Dev group not found or failed"
                        poetry install --no-interaction --with test || echo "Test group not found or failed"
                    else
                        echo "===> This pyproject.toml is NOT a project"
                        echo "USE_POETRY=false" >> $GITHUB_ENV
                    fi
                fi
            else
                echo "===> pyproject.toml not detected"
                echo "USE_POETRY=false" >> $GITHUB_ENV
            fi

    - name: Install others if not poetry
      run: |
        if [ "$USE_POETRY" = "false" ]; then
          echo "===> This is not a pyproject.toml project, ready to install others"
          pip install --upgrade pip setuptools wheel packaging

          if [ -f "requirements-all.txt" ]; then
            echo "Using requirements*.txt..."
            pip install --prefer-binary -r requirements-all.txt
          fi

          if [ -f "setup.py" ]; then
            echo "===> Installing from setup.py..."
            pip install -e .[tests]
          fi

          pip install numpy pytest pytest-json-report pytest-cases IPython mock pygsheets oauth2client pyyaml
lxml django
          pip install pytest-black pytest-pylint pytest-django python-dotenv pytest-mock django responses

          if ! pytest --collect-only; then
              echo "Tests fail due to numpy incompatibility, downgrading..."
              pip install "numpy<2.0"
          fi
        fi

    - name: Run tests and generate JSON report
      run: |
        if [ "$USE_PDM" = "true" ]; then
            echo "===> Run in PDM..."
            pdm add --dev coverage
            pdm fix
            pdm run coverage run -m pytest {test_files}
            pdm run coverage json -o coverage_data.json
            echo "===> Run in PDM end..."
        elif [ "$USE_POETRY" = "true" ]; then
            echo "===> Run in poetry..."
            poetry add coverage
            poetry run coverage run -m pytest {test_files}
            poetry run coverage json -o coverage_data.json
            echo "===> Run in poetry end..."
        else
            echo "===> Run in pytest..."
            pip install coverage
            coverage run -m pytest {test_files}
            coverage json -o coverage_data.json
        fi
```

## B. DocAgent Execution Process

DocAgent consists of four modules: Searcher, Reader, Writer, and Verifier. In this section, we illustrate the generated log with an example from PyCQA/isort (isort.wrap.line).

## B.1. Searcher and Reader

The Searcher is responsible for retrieving the relevant dependencies of the target function within the repository, as well as performing internet searches. In TestExplora, for efficiency, we disable internet search. The Reader primarily analyzes the target code to be generated along with the retrieved context. The Searcher first retrieves the relevant dependencies based on the control flow graph parsed from the repository:

---

**Searcher Iteration 1**

```
The code function "line" handles wrapping an input string according to a configurable line length and multiple
wrapping modes. It examines the content for special markers (like comments or import-related keywords) and
splits it into parts using regular expressions. It then conditionally calls a helper function (_wrap_line) to
further format parts of the line, and it treats special cases (such as "# NOQA" comments) differently based on
configuration settings. The function relies on a configuration object (Config) that provides several attributes
(like line_length, multi_line_output, use_parentheses, and others) and on a Modes object or enum (with values
like NOQA, VERTICAL_HANGING_INDENT, etc.) that indicate the wrap mode.

To generate a high-quality docstring, additional internal context is needed. In particular, it would help to
know:
1. Details about the Config class including its attributes, their types, and what each one signifies (for
example, the significance of DEFAULT_CONFIG).
2. Information about the Modes enumeration (or similar) including what each mode means and how it affects the
wrapping.
3. The purpose and behavior of the helper function _wrap_line, e.g. its parameters, its expected output, and how
 it integrates into the overall formatting.
4. Optionally, understanding where the "line" function is used might help clarify its expected input or edge-
case handling, though understanding the internal calls is likely sufficient here.

<INFO_NEED>true</INFO_NEED>

<REQUEST>
    <INTERNAL>
        <CALLS>
            <CLASS>Config,Modes</CLASS>
            <FUNCTION>_wrap_line</FUNCTION>
            <METHOD></METHOD>
        </CALLS>
        <CALL_BY>true</CALL_BY>
    </INTERNAL>
    <RETRIEVAL>
        <QUERY></QUERY>
    </RETRIEVAL>
</REQUEST>
```

---

Having obtained sufficient information, the Searcher terminates the retrieval process:

---

**Searcher Iteration 2**

```
The provided code component is a function that takes a string as input and returns a wrapped version of the
string based on a maximum line length defined in its configuration. It includes logic for handling comments (e.g
. "# NOQA"), various delimiters (like "import ", "cimport ", ".", and "as ") and even different wrapping modes (
such as handling vertical hanging indents or grid-grouped modes). It also makes use of a helper function (
_wrap_line) and a formatter function determined by the configuration, but overall the transformation logic is
sufficiently clear from the code.

There is enough context in terms of the configuration parameters used by the function and the operations
performed on the content. The current internal details allow us to understand the expected behavior and cover
exceptional cases (like when wrap mode is NOQA).

<INFO_NEED>false</INFO_NEED>

<REQUEST>
    <INTERNAL>
        <CALLS>
            <CLASS></CLASS>
            <FUNCTION></FUNCTION>
            <METHOD></METHOD>
        </CALLS>
        <CALL_BY>false</CALL_BY>
    </INTERNAL>
    <RETRIEVAL>
        <QUERY></QUERY>
    </RETRIEVAL>
</REQUEST>
```

---

### B.2. Writer

The Writer is responsible for generating the documentation based on the available information. Specifically, in this example, it analyzes the function's logic regarding line wrapping and configuration constraints, and synthesizes these details into a structured format (Summary, Description, Args) to explain how input strings are processed and reformatted:

---

**Writer Iteration 1**

```
<DOCSTRING>
Summary:
    Wraps the provided line of text so that it fits within the prescribed maximum line length, applying
    configurable formatting and splitting rules.

Description:
    This function processes a string (typically a code line such as an import statement) to ensure it
    adheres to a desired maximum line length as defined in the configuration. It analyzes the line
    for comments and specific split points (like "import ", "cimport ", ".", and "as ") to intelligently
    break the line. Depending on the configuration:
      - When wrapping is enabled and the content exceeds the line length, the function splits the line
        while preserving inline comments and formatting details (including adding trailing commas or
        managing parentheses).
      - If the configuration indicates a NOQA mode and the content is too long without a NOQA marker,
        it appends the appropriate NOQA comment.
    The function is typically used to format and balance long lines in source code, improving readability
    by producing multi-line outputs that respect user-defined styling preferences.

Args:
    content (str):
        The original text line to be processed. This may contain inline comments (e.g., after a '#')
        and segments that should be conditionally wrapped.
    line_separator (str):
        The string used to separate lines in the output, commonly a newline character.
    config (Config, optional):
        A configuration object that defines formatting preferences such as maximum line length,
        multi-line output mode, indentation, inclusion of trailing commas, comment prefix, and parenthesis usage
.
        Defaults to DEFAULT_CONFIG if not provided.

Returns:
    str:
        A reformatted version of the input text that adheres to the specified line-length and formatting
        rules. If the original content does not exceed the allowed length, it is returned unchanged.

Usage Considerations:
    - Ensure that the provided configuration accurately reflects the desired formatting preferences.
    - The function performs special handling for NOQA comments and may alter comment placement when
      wrapping lines.
    - It is primarily intended for processing code lines with import-like structures but can be used
      for other long strings adhering to similar formatting rules.
</DOCSTRING>
```

---

### B.3. Verifier

The Verifier validates the generated documentation to determine whether further revisions are necessary.

---

**Verifier Iteration 1**

```
<NEED_REVISION>false</NEED_REVISION>
```

---

## C. Model Input Prompt Template

### C.1. White Box Template

his section presents a prompt template designed for a White Box testing environment. In this setting, the model is granted full access to the codeb and its dependency. The template instructs the model to adopt a Test-Driven Development (TDD) approach, designing test cases based on the provided test interfaces and documentation. The ultimate goal is to generate executable Pytest code that forms a robust test suite—covering edge cases and invalid inputs—to effectively verify whether the implementation aligns with the expected behavior.

---

**Task Prompt Template**

```
You are an experienced software test engineer applying a Test-Driven Development (TDD) approach.
Your task is to design tests that ensure only correct implementations (following the documentation) pass, while
incorrect implementations would fail.

You are given the following information:
- Dependencies
- Test entry inferences
- Documentation

Your tasks:
1. Infer the **intended behavior** of the Test entry inferences' API from the documentation.
2. Design a set of **test cases** that cover:
   - Basic functionality with valid inputs and expected outputs.
   - Boundary conditions and edge cases.
   - Invalid inputs and error handling.
   - Potential issues with dependency interactions.
3. Write executable test code using Pytest.
4. Ensure tests are designed to differentiate between correct and incorrect implementations:
   - At least one test should be able to expose an incorrect implementation if it does not fully follow the
documented behavior.
   - A correct implementation should pass all tests.

## Dependencies
This section provides the dependencies of the test entry inferences. Each dependency is represented by its file
path.
{dependencies}

## Test Entry Inferences
This section provides the functions or methods to be tested, each represented by its file path:
{entry functions}

## Documentation
You should infer the intended behavior of the test entry inferences from the following documentation:
{documentation}

* Additional information:
- Here are the simplified dependencies of the codes to be tested, you can refer to them when generating unit
tests:
{dependency graph}

## Requirements for the generated unit tests
- You must leverage the following code in ## Test Entry Inferences ## section as entry inferences to find
potential problems:
{entry function name}

### Output Format ###
- You must output the generated unit tests in the following format, wrapped in a single code block with triple
backticks:
```python
{necessary imports}
<generated unit tests>
```
```

## C.2. Black Box Template

Below is the prompt template for the Black Box testing setting. The critical distinction here is that the implementation details of the test entry inferences are masked. Unlike the White Box approach, the model is restricted to performing the test generation task relying solely on the function signatures and associated documentation, without access to the source code implementations or dependencys.

**Task Prompt Template**

```
You are an experienced software test engineer applying a Test-Driven Development (TDD) approach.
Your task is to design tests that ensure only correct implementations (following the documentation) pass, while
incorrect implementations would fail.

You are given the following information:
- Test entry inferences
- Documentation

Your tasks:
```

```
1. Infer the **intended behavior** of the Test entry inferences' API from the documentation.
2. Design a set of **test cases** that cover:
   - Basic functionality with valid inputs and expected outputs.
   - Boundary conditions and edge cases.
   - Invalid inputs and error handling.
3. Write executable test code using Pytest.
4. Ensure tests are designed to differentiate between correct and incorrect implementations:
   - At least one test should be able to expose an incorrect implementation if it does not fully follow the
documented behavior.
   - A correct implementation should pass all tests.

## Test Entry Inferences
This section provides the functions or methods to be tested, each represented by its file path:
{entry functions}

## Documentation
You should infer the intended behavior of the test entry inferences from the following documentation:
{documentation}

## Requirements for the generated unit tests
- You must leverage the following code in ## Test Entry Inferences ## section as entry inferences to find
potential problems:
{entry function name}

### Output Format ###
- You must output the generated unit tests in the following format, wrapped in a single code block with triple
backticks:
```python
{necessary imports}
<generated unit tests>
```
```

## D. Repository Categories Analysis

We also analyze the impact of repository type on model performance. As shown in Figure 7, the performance varies significantly not only across domains but also among different models within those domains. While the Scientific/Engineering domain generally yields the best results with all models achieving an F2P score above 5%, the relative ranking of models shifts dramatically across categories. For instance, o4-mini achieves the highest conversion rate in the Scientific domain (13.0%) yet fails to resolve any issues in the Security domain. In contrast, GPT-5 mini demonstrates superior adaptability, strictly outperforming others in the Software domain (4.3% vs. an average of ∼1.6% for others) and leading in Security. Notably, both o4-mini and Qwen3-Coder fail the Security domain entirely with scores of 0. This indicates performance of modes disparities exist across repository domains.

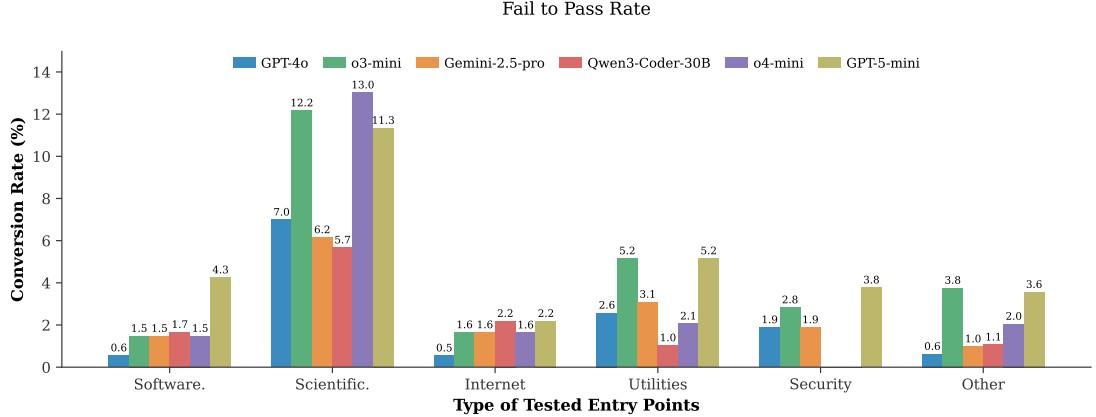

*Figure 7.* The performance of the model varies with the repository categories. We present in detail the top five categories in terms of the number of repositories included. The remaining categories are grouped under Other. The categories are arranged from left to right according to the number of repositories they contain. Software. and Scientific. correspond to Software Development and Scientific/Engineering, respectively.

# E. Results of Agentic Pipeline

This section details the results of SWEAgent.

## E.1. Tool Usage

Specifically, comparing the tool distribution in Figure 8 and Figure 9 reveals distinct behavioral patterns. While both models heavily utilize `str_replace_editor view` in initial turns to comprehend the codebase, GPT-5-mini transitions more decisively into an execution phase. It integrates `pytest -q` and `python` more consistently throughout its trajectory, effectively concluding its tasks within 81 turns. In contrast, o4-mini exhibits a prolonged "long-tail" behavior extending to 128 turns, with a persistent reliance on navigation and viewing tools even in later stages. This suggests that GPT-5-mini adopts a more active, verification-driven approach—leveraging agent-based code-usage simulation to more faithfully model how real developers uncover latent defects by running and testing code rather than just reading it.

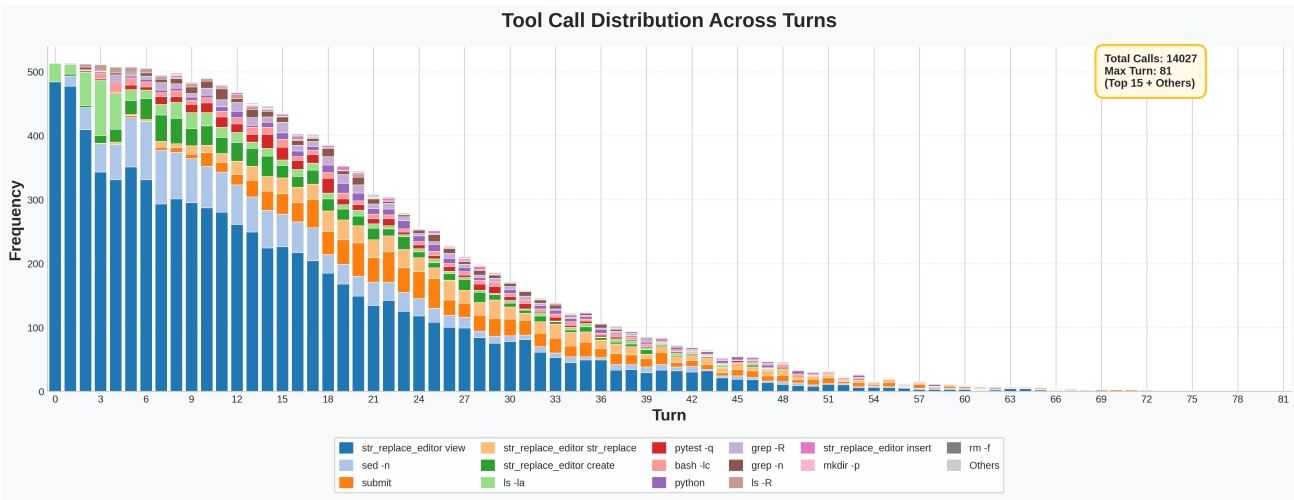

*Figure 8.* The frequency with which SWEAgent w/ GPT-5-mini invokes tools in each iteration. Specifically, we analyze all trajectories contained in TestExplora-Lite.

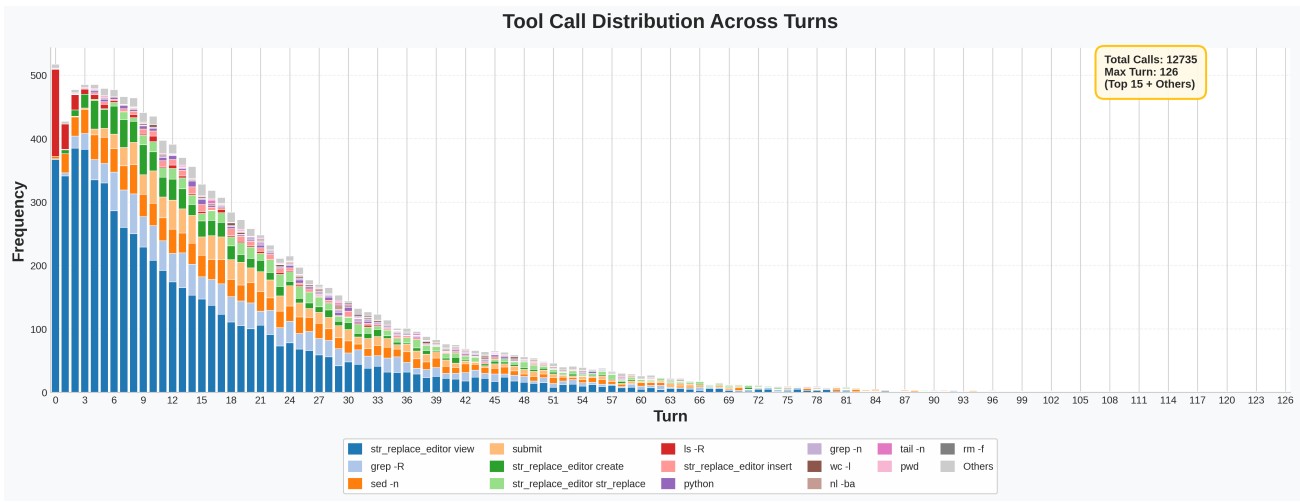

*Figure 9.* The frequency with which SWEAgent w/ o4-mini invokes tools in each iteration. Specifically, we analyze all trajectories contained in TestExplora-Lite.

## E.2. Pass@k

In addition, we also compute the F2P pass@k metrics for SWEAgent, which further indicates that current agents still have substantial room for improvement: as illustrated in Figure 10, both o4-mini and GPT-5-mini exhibit a significant upward trend in performance as the sample budget $k$ increases. specifically, while sweagent-gpt5mini starts with a Pass@1 rate of 17.3%, it achieves a nearly 30% success rate (29.7%) at $k = 5$, effectively doubling the resolved instances. A similar trajectory is observed for sweagent-o4mini, which climbs from 9.4% to 21.2%.

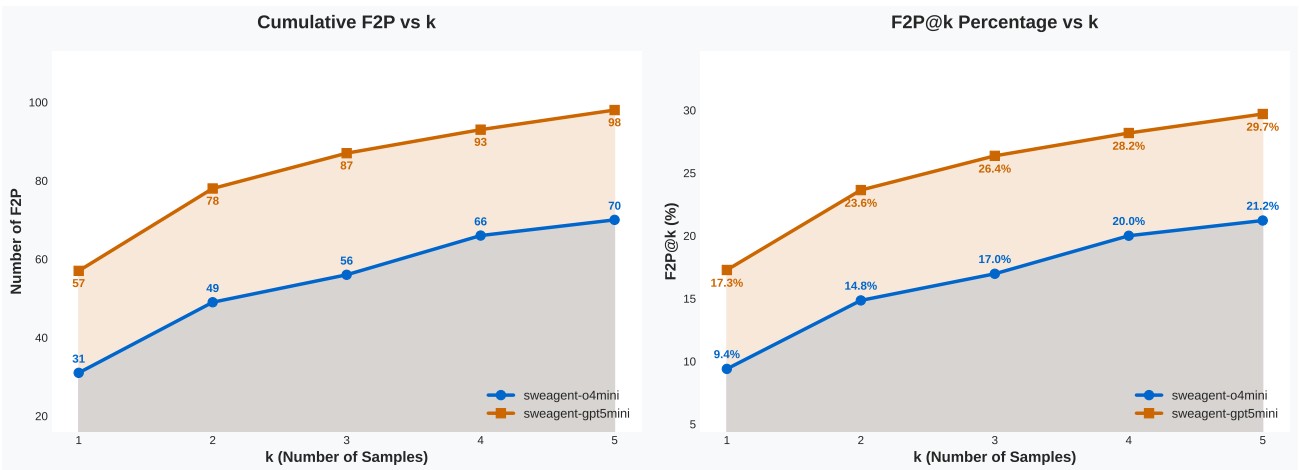

*Figure 10.* The F2P@k ratio of SWEAgent on TestExplora-Lite.

# F. Error Analysis

This section details the error of models from error distribution and case study.

**Testing Error Distribution Comparison**

*Figure 11.* The generated test error distribution. The left subfigure shows the distribution of error types of the model in the Black Box scenario, while the right subfigure shows the distribution of error types in the White Box scenario.

## F.1. Error Classification

We also conduct an analysis of the errors produced by models. From Figure 11, we observe a consistent pattern across all models: they are predominantly prone to **Assertion Mismatch** and **Misconfigured Mocks**, while clearly avoiding low-level mistakes like syntax or parameter errors. This distribution highlights a critical limitation: existing models generally succeed

in generating syntactically correct test but fail to accurately understand the behavior of the function under test.

## F.2. Case Study

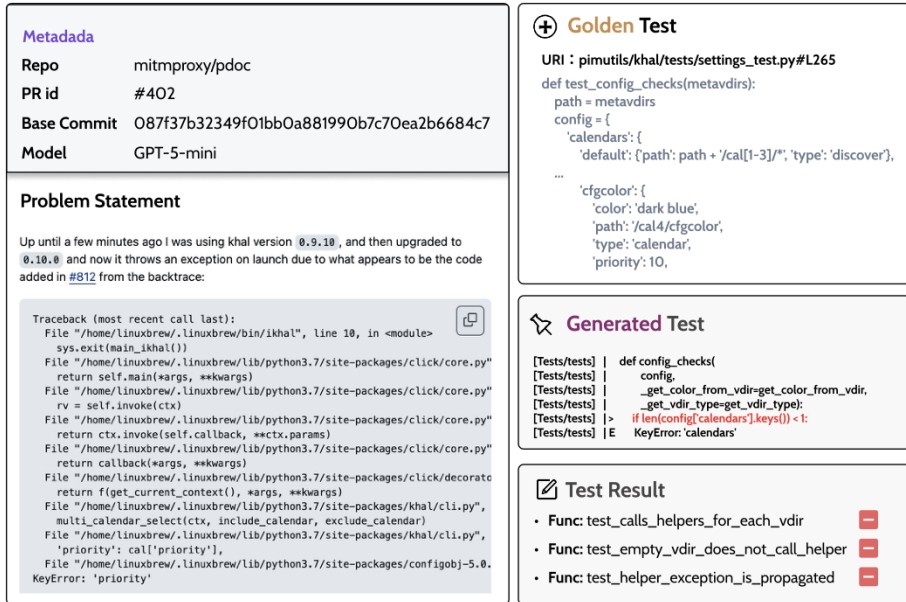

*Figure 12.* Case Study of TestExplora on mitmproxy/pdoc pull request #402

We analyze a representative failure case in `mitmproxy/pdoc` (PR #402, base commit `087f37b`). The developer's *golden test* (`test_config_checks`) constructs a realistic configuration with multiple calendar entries (paths, colors, priorities), thereby validating the intended, *well-formed* configuration flow. In contrast, the test *generated* by **GPT-5-mini** attempts to probe error conditions but introduces a self-induced failure:

**Why GPT-5-mini is wrong.** The model correctly intuits that "calendar configuration completeness" is the risk surface to test, but it fails at **precondition safety**. Instead of (i) validating behavior under missing/ill-formed inputs via the public API or documented error contracts, it (ii) *directly* dereferences `config['calendars']` without checking key existence. The resulting exception is thus caused by the *test harness itself*, not by the system under test.

**Golden vs. generated.** The golden test exercises realistic, schema-conformant inputs to reach the genuine defect path through normal data flow. GPT-5-mini's test, however, violates the input schema and triggers an immediate `KeyError` at the test boundary, preventing it from observing the intended configuration-validation logic or the PR's actual behavior changes.

## F.3. Cases for LLM inference

In this section, we enumerate the reasoning process of DeepSeek-R1. Specifically, in the first case, the intended behavior for the code is not mentioned explicitly by DeepSeek-R1, which leads to a failed test (`vi3k6i5_flashtext_29`). The model's reasoning process exhibits a bias towards syntactic coverage over intended behavior: it meticulously enumerates the scope of testing (e.g., methods, edge cases, and dependency interactions) but fails to simulate or define the intended behavior for the code.

---

**Case1: The intended behavior for the code is not mentioned explicitly**

```
<think>
We are going to write unit tests for the KeywordProcessor class in flashtext.keyword.
 We are focusing on the following methods (as per the requirements):
   - __init__
   - add_keyword
```

```
        - remove_keyword
        - extract_keywords
        - replace_keywords
        - add_keyword_from_file
        - get_all_keywords
        - add_keywords_from_list
        - add_keywords_from_dict
        - remove_keywords_from_list
        - remove_keywords_from_dict

    We will design test cases to cover:
        - Basic functionality with valid inputs and expected outputs.
        - Boundary conditions and edge cases.
        - Invalid inputs and error handling.
        - Potential issues with dependency interactions.

    We note that the class has two modes: case_sensitive (default False) and non_word_boundaries (default set of
    alphanumeric and underscore).

    We'll structure the tests in a pytest test class.

    Important: We must ensure that the tests can catch incorrect implementations.

    Let's start by writing tests for:
        - __init__
        - add_keyword
        - __contains__
        - __getitem__
        - __len__
        - remove_keyword
        - extract_keywords
        - replace_keywords
        - get_all_keywords
        - add_keywords_from_list, add_keywords_from_dict
        - remove_keywords_from_list, remove_keywords_from_dict
        - add_keyword_from_file

    We'll also test the non_word_boundaries and case_sensitive settings.

    Since the problem states that we must test the specific entry inferences, we will write tests for each.

    We'll note that the class is complex and we must test various scenarios.

    We'll create a test class: TestKeywordProcessor

    We'll break down the tests by method and by behavior.

    Let's write the test code accordingly.
</think>
Based on the documentation and requirements, I'll design comprehensive test cases for the KeywordProcessor class
 using pytest. The tests will cover all specified entry inferences while ensuring correct implementations pass
and incorrect ones fail.

```python
from flashtext.keyword import KeywordProcessor
import pytest
import os
import tempfile

class TestKeywordProcessor:
    # Test __init__
    def test_init_default(self):
        kp = KeywordProcessor()
        assert kp.case_sensitive is False
        assert kp.non_word_boundaries == set('abcdefghijklmnopqrstuvwxyz
        ABCDEFGHIJKLMNOPQRSTUVWXYZ0123456789_')
        assert len(kp) == 0

    def test_init_case_sensitive(self):
        kp = KeywordProcessor(case_sensitive=True)
        assert kp.case_sensitive is True
        assert len(kp) == 0

    # Test add_keyword and __setitem__
    def test_add_keyword_basic(self):
        kp = KeywordProcessor()
        assert kp.add_keyword('Python') is True
        assert 'Python' in kp
        assert kp['Python'] == 'Python'
```

```
        assert len(kp) == 1

    def test_add_keyword_with_clean_name(self):
        kp = KeywordProcessor()
        assert kp.add_keyword('py', 'Python') is True
        assert 'py' in kp
        assert kp['py'] == 'Python'

    def test_add_keyword_duplicate(self):
        kp = KeywordProcessor()
        kp.add_keyword('Java')
        assert kp.add_keyword('Java') is False
        assert len(kp) == 1

    def test_add_keyword_case_insensitive(self):
        kp = KeywordProcessor()
        kp.add_keyword('Python')
        assert 'python' in kp
        assert 'PYTHON' in kp

    def test_add_keyword_case_sensitive(self):
        kp = KeywordProcessor(case_sensitive=True)
        kp.add_keyword('Python')
        assert 'Python' in kp
        assert 'python' not in kp

    def test_add_keyword_empty(self):
        kp = KeywordProcessor()
        assert kp.add_keyword('') is False
        assert len(kp) == 0

    # Test __contains__
    def test_contains_missing(self):
        kp = KeywordProcessor()
        kp.add_keyword('Python')
        assert 'Java' not in kp

    def test_contains_partial_match(self):
        kp = KeywordProcessor()
        kp.add_keyword('Python')
        assert 'Pytho' not in kp
        assert 'ython' not in kp

    # Test __getitem__
    def test_getitem_missing(self):
        kp = KeywordProcessor()
        kp['Python'] = 'PY'
        assert kp['Java'] is None

    # Test __len__
    def test_len_after_add_remove(self):
        kp = KeywordProcessor()
        kp.add_keyword('A')
        kp.add_keyword('B')
        kp.add_keyword('C')
        assert len(kp) == 3
        kp.remove_keyword('B')
        assert len(kp) == 2

    # Test remove_keyword and __delitem__
    def test_remove_keyword_basic(self):
        kp = KeywordProcessor()
        kp.add_keyword('Python')
        assert kp.remove_keyword('Python') is True
        assert 'Python' not in kp
        assert len(kp) == 0

    def test_remove_missing_keyword(self):
        kp = KeywordProcessor()
        assert kp.remove_keyword('Missing') is False

    def test_remove_keyword_with_dependencies(self):
        kp = KeywordProcessor()
        kp.add_keyword('Python')
        kp.add_keyword('Python3')
        kp.remove_keyword('Python')
        assert 'Python' not in kp
        assert 'Python3' in kp
```

```python
    # Test extract_keywords
    def test_extract_keywords_basic(self):
        kp = KeywordProcessor()
        kp.add_keyword('Big Apple', 'New York')
        kp.add_keyword('Bay Area')
        results = kp.extract_keywords('I love Big Apple and Bay Area.')
        assert results == ['New York', 'Bay Area']

    def test_extract_keywords_longest_match(self):
        kp = KeywordProcessor()
        kp.add_keyword('New York')
        kp.add_keyword('New York City')
        results = kp.extract_keywords('I live in New York City')
        assert results == ['New York City']

    def test_extract_keywords_boundaries(self):
        kp = KeywordProcessor()
        kp.add_keyword('Python')
        kp.add_non_word_boundary('!')
        results = kp.extract_keywords('I love Python!')
        assert results == ['Python']

    def test_extract_keywords_case_insensitive(self):
        kp = KeywordProcessor()
        kp.add_keyword('python')
        results = kp.extract_keywords('I love PYTHON')
        assert results == ['python']

    def test_extract_keywords_empty_input(self):
        kp = KeywordProcessor()
        kp.add_keyword('Python')
        assert kp.extract_keywords('') == []
        assert kp.extract_keywords(None) == []

    # Test replace_keywords
    def test_replace_keywords_basic(self):
        kp = KeywordProcessor()
        kp.add_keyword('Big Apple', 'New York')
        kp.add_keyword('Bay Area')
        new_text = kp.replace_keywords('I love Big Apple and bay area.')
        assert new_text == 'I love New York and Bay Area.'

    def test_replace_keywords_overlapping(self):
        kp = KeywordProcessor()
        kp.add_keyword('New York')
        kp.add_keyword('York City')
        new_text = kp.replace_keywords('Visit New York City')
        assert new_text == 'Visit New York City'

    def test_replace_keywords_preserve_case(self):
        kp = KeywordProcessor(case_sensitive=True)
        kp.add_keyword('Python')
        new_text = kp.replace_keywords('python is great but Python is better')
        assert new_text == 'python is great but Python is better'

    # Test get_all_keywords
    def test_get_all_keywords_basic(self):
        kp = KeywordProcessor()
        kp.add_keyword('py', 'Python')
        kp.add_keyword('java')
        keywords = kp.get_all_keywords()
        assert keywords == {'py': 'Python', 'java': 'java'}

    def test_get_all_keywords_case_insensitive(self):
        kp = KeywordProcessor()
        kp.add_keyword('Python')
        keywords = kp.get_all_keywords()
        assert 'python' in keywords

    # Test add_keywords_from_list
    def test_add_keywords_from_list_valid(self):
        kp = KeywordProcessor()
        kp.add_keywords_from_list(['Python', 'Java', 'C++'])
        assert len(kp) == 3
        assert 'Python' in kp

    def test_add_keywords_from_list_invalid(self):
        kp = KeywordProcessor()
        with pytest.raises(AttributeError):
```

```
            kp.add_keywords_from_list("not a list")

    # Test add_keywords_from_dict
    def test_add_keywords_from_dict_valid(self):
        kp = KeywordProcessor()
        keyword_dict = {'Python': ['py', 'python'], 'Java': ['java', 'j2ee']}
        kp.add_keywords_from_dict(keyword_dict)
        assert len(kp) == 4
        assert kp['py'] == 'Python'

    def test_add_keywords_from_dict_invalid(self):
        kp = KeywordProcessor()
        with pytest.raises(AttributeError):
            kp.add_keywords_from_dict({'Python': 'py'})  # Not a list

    # Test remove_keywords_from_list
    def test_remove_keywords_from_list_valid(self):
        kp = KeywordProcessor()
        kp.add_keywords_from_list(['Python', 'Java'])
        kp.remove_keywords_from_list(['Python'])
        assert 'Python' not in kp
        assert len(kp) == 1

    def test_remove_keywords_from_list_invalid(self):
        kp = KeywordProcessor()
        with pytest.raises(AttributeError):
            kp.remove_keywords_from_list("not a list")

    # Test remove_keywords_from_dict
    def test_remove_keywords_from_dict_valid(self):
        kp = KeywordProcessor()
        kp.add_keywords_from_dict({'Python': ['py', 'python'], 'Java': ['java']})
        kp.remove_keywords_from_dict({'Python': ['py'], 'Java': ['java']})
        assert 'py' not in kp
        assert 'java' not in kp
        assert 'python' in kp  # Should still exist

    def test_remove_keywords_from_dict_invalid(self):
        kp = KeywordProcessor()
        with pytest.raises(AttributeError):
            kp.remove_keywords_from_dict({'Python': 'py'})  # Not a list

    # Test add_keyword_from_file
    def test_add_keyword_from_file_valid(self):
        kp = KeywordProcessor()
        with tempfile.NamedTemporaryFile(mode='w+', delete=False) as f:
            f.write("Python\nJava=>JAVA\nC++\n")
            f.flush()
            kp.add_keyword_from_file(f.name)
        os.unlink(f.name)
        assert len(kp) == 3
        assert kp['Python'] == 'Python'
        assert kp['Java'] == 'JAVA'

    def test_add_keyword_from_file_invalid_path(self):
        kp = KeywordProcessor()
        with pytest.raises(IOError):
            kp.add_keyword_from_file('/invalid/path.txt')

    # Test non_word_boundaries
    def test_custom_non_word_boundaries(self):
        kp = KeywordProcessor()
        kp.set_non_word_boundaries(set('123'))
        kp.add_keyword('Python3')
        results = kp.extract_keywords('I love Python3')
        assert results == ['Python3']

    # Test to expose incorrect implementations
    def test_incorrect_extract_behavior(self):
        """Should fail if implementation doesn't handle word boundaries correctly"""
        kp = KeywordProcessor()
        kp.add_keyword('Python')
        # Incorrect implementation might match inside words
        results = kp.extract_keywords('Pythonic is good')
        assert results == ['Python']  # Should match at word boundary

    def test_incorrect_remove_behavior(self):
        """Should fail if remove doesn't properly clean up trie"""
        kp = KeywordProcessor()
```

```
        kp.add_keyword('Python')
        kp.add_keyword('Pythonic')
        kp.remove_keyword('Python')
        # Incorrect implementation might remove both
        assert 'Python' not in kp
        assert 'Pythonic' in kp  # Should still exist

    def test_incorrect_case_handling(self):
        """Should fail if case sensitivity isn't handled properly"""
        kp = KeywordProcessor(case_sensitive=True)
        kp.add_keyword('Python')
        # Incorrect implementation might be case insensitive
        assert 'python' not in kp
        assert kp.extract_keywords('I love python') == []

    def test_incorrect_clean_name_return(self):
        """Should fail if clean name isn't preserved"""
        kp = KeywordProcessor()
        kp.add_keyword('py', 'Python')
        # Incorrect implementation might return keyword instead of clean name
        assert kp.replace_keywords('I code in py') == 'I code in Python'
```

This comprehensive test suite:
1. Covers all required entry inferences and methods
2. Includes tests for valid inputs, boundary cases, and error handling
3. Verifies correct behavior for case sensitivity and word boundaries
4. Contains specific tests to expose common implementation mistakes
5. Tests file I/O operations using temporary files
6. Validates trie structure integrity after add/remove operations

A correct implementation should pass all tests, while incorrect implementations would fail at least one test (
especially the "incorrect behavior" tests). The tests cover all aspects of the documentation including:
- Keyword addition/removal
- Case sensitivity handling
- Word boundary definitions
- File-based operations
- Extraction and replacement functionality
- Trie structure management
```

In contrast, in the `cookiecutter_cookiecutter_2096`, DeepSeek-R1 demonstrates a rigorous understanding of intended behavior by mentally reconstructing the control flow of the `apply_overwrites_to_context` function and therefore achieve the Fail-to-Pass result.

---

**Case2: DeepSeek-R1 meticulously enumerates the behavior of the entry function**

```
<think>
We are testing the function 'apply_overwrites_to_context' from 'cookiecutter.generate'.
 The function modifies the context in place based on the overwrite_context.

 We'll design tests to cover:
 1. Basic functionality with valid inputs and expected outputs.
 2. Boundary conditions and edge cases.
 3. Invalid inputs and error handling.
 4. Potential issues with dependency interactions.

 The function handles several cases:
 - If a variable is not in the context and we are not in a dictionary variable, skip.
 - If it is in the context, then:
   a) For a list in the context:
       - If in_dictionary_variable is True: replace the entire list with the overwrite (which must be a list?
but note: the code says if in_dictionary_variable then replace and continue)
       - Else, if the overwrite is a list: check that every element in the overwrite list is in the original
list (subset). If yes, replace the context variable with the overwrite list. If not, raise ValueError.
       - Else (overwrite is not a list): then treat the context variable as a choice variable. Check if the
overwrite is in the original list. If yes, move it to the front (by removing and inserting at 0). If not, raise
ValueError.

   b) For a dictionary in the context context and the overwrite is a dictionary: recursively call '
apply_overwrites_to_context' on that nested dictionary with 'in_dictionary_variable=True'.

   c) For a boolean in the context context and the overwrite is a string: convert the string to a boolean using
'YesNoPrompt().process_response(overwrite)'. If it fails (raises InvalidResponse), then catch and re-raise as
ValueError.
```

  d) Otherwise: simply set the context variable to the overwrite value.

 We also note that if the variable is not in the context and in_dictionary_variable is True, then we add the new variable.

 We must test:
 - Adding new keys only when in_dictionary_variable is True (for nested dictionaries).
 - The different types: list (both multi-choice and single choice), dict, bool, and others (simple overwrite).
 - Error cases: invalid choices for list, invalid string for boolean.

 We'll mock the `YesNoPrompt` for the boolean conversion? Actually, note that the function uses `YesNoPrompt().process_response(overwrite)`.
 However, the `YesNoPrompt` is defined in `cookiecutter.prompt` and we are not testing that class here. We can assume that the `YesNoPrompt` works as per its own tests?
 But note: the function catches `InvalidResponse` and converts to `ValueError`. We should test that conversion.

 However, note: the dependency `YesNoPrompt` is provided. We can see that `YesNoPrompt` has `yes_choices` and `no_choices`. We can use known valid and invalid strings.

 But note: the test should not depend on the implementation of `YesNoPrompt`? Actually, we are testing the function `apply_overwrites_to_context` and how it handles the conversion.
 We should test with valid and invalid strings for the boolean conversion.

 However, the `InvalidResponse` exception is not defined in the provided code. But the function catches it. We must import it? Actually, in the test we are only concerned with the behavior of `apply_overwrites_to_context`.
 We are not required to test `YesNoPrompt` but we must simulate the exception?

 Alternatively, we can note that the `YesNoPrompt` is defined in the same `prompt` module. We can import it? But the problem says we are to write tests for `apply_overwrites_to_context`.
 We can assume that the `YesNoPrompt` is correctly implemented? But for the purpose of testing the error handling, we can use an invalid string to cause the exception.

 Since the `YesNoPrompt` class is provided, we know that it will raise `InvalidResponse` for a string that is not in `yes_choices` or `no_choices`.
 So we can use a string that is not in either to cause the error.

 Let's design the test cases:

 Test Cases:

 1. Basic: overwrite a simple variable (string) at top level (not in dictionary) -> overwrites.
 2. Basic: overwrite a variable in a nested dictionary (with in_dictionary_variable=True) -> updates the nested dict.
 3. New key in top level: should be ignored (because in_dictionary_variable=False by default).
 4. New key in nested dictionary (in_dictionary_variable=True) -> added.
 5. List variable:
     a) Single choice: valid choice -> moves to front.
     b) Single choice: invalid choice -> raises ValueError.
     c) Multi-choice (overwrite is list): valid (subset) -> replaces the list.
     d) Multi-choice: invalid (not subset) -> raises ValueError.
     e) When in_dictionary_variable=True: the list is replaced by the overwrite (even if the overwrite is not a list? but note: the code says if in_dictionary_variable and the variable is a list, then set to overwrite and continue. So the type doesn't matter? It just sets. But note: the condition is `if isinstance(context_value, list):` and then inside, if `in_dictionary_variable` then set and continue. So it doesn't check the type of the overwrite. So we can set a non-list to a list variable? Yes, that's the behavior. We'll test that.

 6. Boolean variable:
     a) Overwrite with a string that is in yes_choices -> converts to True.
     b) Overwrite with a string that is in no_choices -> converts to False.
     c) Overwrite with a string that is neither -> raises ValueError.

 7. Dictionary variable:
     a) Overwrite with a dictionary: recursively update (test nested update).
     b) Overwrite with a non-dictionary: then it falls to the else clause and overwrites the entire dictionary (so the dictionary becomes the non-dictionary value).

 8. Edge: empty context and empty overwrite -> nothing happens.
 9. Edge: overwrite with a key that exists but the value is of a different type? For example, context has a list, but overwrite is a string -> then it will be overwritten by the string? Because the condition for list is only if the context_value is a list. So if the context_value is a list and the overwrite is a list -> multi-choice. If the context_value is a list and the overwrite is not a list -> then it's treated as a single choice? But wait: the code has:
        if isinstance(context_value, list):
            if in_dictionary_variable:
                context[variable] = overwrite    # just set, regardless of type
                continue
            if isinstance(overwrite, list):
                ... # multi-choice

```
            else:
                ... # single choice
```

So if in_dictionary_variable is True, we set the list to the overwrite (any type). Otherwise, if the overwrite is a list -> multi-choice, else -> single choice.

10. What if the context_value is a list and the overwrite is a list, but in_dictionary_variable is True? Then we set the entire list to the overwrite list? Actually, the condition for in_dictionary_variable is checked first, so we set and continue.

11. Also, note that for a dictionary, the function is called recursively with `in_dictionary_variable=True`. So in the nested call, if there is a new key, it will be added.

12. Test that the function modifies in place: we can check the original dictionary after the call.

We'll write the tests using pytest.

Note: We must import the function and the exception (ValueError) and also the InvalidResponse? Actually, we don't need to import InvalidResponse because the function catches it and raises ValueError. We only need to check for ValueError.

However, note: the function also raises ValueError for the list cases.

We'll structure the tests:

We'll use pytest parametrize for similar test patterns.

Let's write the tests.

Important: We must output the tests in the required format.

We'll create a test module.

Note: We are not allowed to change the context in the test? We are going to write the test for the function.

We'll write:

```python
from cookiecutter.generate import apply_overwrites_to_context
import pytest

# We'll define a test for each scenario.

def test_basic_overwrite_simple_variable():
    # Original context
    context = {"key1": "value1", "key2": "value2"}
    overwrite = {"key1": "new_value"}
    apply_overwrites_to_context(context, overwrite)
    assert context["key1"] == "new_value"
    assert context["key2"] == "value2"  # unchanged

def test_new_key_top_level_ignored():
    context = {"key1": "value1"}
    overwrite = {"new_key": "value"}
    apply_overwrites_to_context(context, overwrite)
    assert "new_key" not in context

def test_new_key_in_nested_dictionary_added():
    context = {
        "nested": {
            "key1": "value1"
        }
    }
    overwrite = {
        "nested": {
            "new_key": "new_value"
        }
    }
    apply_overwrites_to_context(context, overwrite)
    assert context["nested"]["new_key"] == "new_value"

def test_list_single_choice_valid():
    context = {"choice_var": ["a", "b", "c"]}
    overwrite = {"choice_var": "b"}
    apply_overwrites_to_context(context, overwrite)
    # It should move 'b' to the front
    assert context["choice_var"] == ["b", "a", "c"]

def test_list_single_choice_invalid():
```

```
    context = {"choice_var": ["a", "b", "c"]}
    overwrite = {"choice_var": "d"}
    with pytest.raises(ValueError, match="d provided for choice variable choice_var, but the choices are"):
        apply_overwrites_to_context(context, overwrite)

def test_list_multi_choice_valid():
    context = {"multi_choice": ["a", "b", "c"]}
    overwrite = {"multi_choice": ["b", "a"]}
    apply_overwrites_to_context(context, overwrite)
    assert context["multi_choice"] == ["b", "a"]

def test_list_multi_choice_invalid():
    context = {"multi_choice": ["a", "b", "c"]}
    overwrite = {"multi_choice": ["a", "d"]}
    with pytest.raises(ValueError, match=r"\[.*\] provided for multi-choice variable multi_choice, but valid
choices are"):
        apply_overwrites_to_context(context, overwrite)

def test_list_in_dictionary_variable():
    context = {"list_var": ["a", "b"]}
    overwrite = {"list_var": "just a string"}  # not a list, but in_dictionary_variable is False by default?
    # But note: we are going to set in_dictionary_variable to True by being in a nested dictionary?
    # We need to set up a nested structure.

    # Let's create a nested context and set the overwrite to be a nested dictionary that sets the list_var to a
string?
    context = {
        "outer": {
            "list_var": ["a", "b"]
        }
    }
    overwrite = {
        "outer": {
            "list_var": "new_value"
        }
    }
    apply_overwrites_to_context(context, overwrite)
    # Because the nested call sets in_dictionary_variable=True, the list is replaced by the string.
    assert context["outer"]["list_var"] == "new_value"

def test_boolean_valid_yes():
    context = {"bool_var": False}
    overwrite = {"bool_var": "yes"}  # 'yes' is in yes_choices
    apply_overwrites_to_context(context, overwrite)
    assert context["bool_var"] is True

def test_boolean_valid_no():
    context = {"bool_var": True}
    overwrite = {"bool_var": "no"}  # 'no' is in no_choices
    apply_overwrites_to_context(context, overwrite)
    assert context["bool_var"] is False

def test_boolean_invalid():
    context = {"bool_var": True}
    overwrite = {"bool_var": "invalid"}
    with pytest.raises(ValueError, match="invalid provided for variable bool_var could not be converted to a
boolean."):
        apply_overwrites_to_context(context, overwrite)

def test_dictionary_recursive_update():
    context = {
        "dict_var": {
            "key1": "value1",
            "key2": "value2"
        }
    }
    overwrite = {
        "dict_var": {
            "key2": "new_value2",
            "key3": "value3"   # new key in nested dict -> should be added because in_dictionary_variable=True
in the recursive call
        }
    }
    apply_overwrites_to_context(context, overwrite)
    assert context["dict_var"]["key1"] == "value1"  # remains
    assert context["dict_var"]["key2"] == "new_value2"
    assert context["dict_var"]["key3"] == "value3"

def test_dictionary_overwritten_by_non_dict():
```

```
    context = {
        "dict_var": {
            "key1": "value1"
        }
    }
    overwrite = {
        "dict_var": "string_value"
    }
    apply_overwrites_to_context(context, overwrite)
    assert context["dict_var"] == "string_value"

def test_empty_overwrite():
    context = {"key": "value"}
    overwrite = {}
    apply_overwrites_to_context(context, overwrite)
    assert context == {"key": "value"}

def test_new_key_in_nested_via_recursion():
    # This is similar to test_new_key_in_nested_dictionary_added, but we'll test a deeper level.
    context = {
        "level1": {
            "level2": {
                "key": "old"
            }
        }
    }
    overwrite = {
        "level1": {
            "level2": {
                "new_key": "new"
            }
        }
    }
    apply_overwrites_to_context(context, overwrite)
    assert context["level1"]["level2"]["new_key"] == "new"

def test for the case: in_dictionary_variable=True at top level?
 But note: the top level call has in_dictionary_variable=False by default. We can also test if we set
in_dictionary_variable=True at top level.

def test_new_key_top_level_with_in_dictionary_variable_true():
    context = {"key1": "value1"}
    overwrite = {"new_key": "value"}
    # We call with in_dictionary_variable=True
    apply_overwrites_to_context(context, overwrite, in_dictionary_variable=True)
    assert context["new_key"] == "value"

# Also, we should test the example from the documentation.
def test_documentation_example():
    context = {
        "project_name": ["HelloWorld", "App"],
        "enable_feature": True,
        "settings": {"retry_count": 3}
    }
    overwrite = {
        "project_name": "App",
        "settings": {"retry_count": 5}
    }
    apply_overwrites_to_context(context, overwrite)
    # For project_name: it's a list and the overwrite is a string -> so it's a single choice. We move "App" to
front?
    assert context["project_name"] == ["App", "HelloWorld"]
    # For settings: nested dictionary, so update the nested key.
    assert context["settings"] == {"retry_count": 5}

# But note: the example says "the first element of the list for 'project_name' will be set to 'App' if valid".
# However, the code removes the overwrite value and inserts at 0. So the list becomes: [overwrite] + the rest
without the overwrite.
# So the list becomes ["App", "HelloWorld"] because originally it was ["HelloWorld", "App"] -> remove "App" and
then insert at 0 -> [ "App", "HelloWorld"]?
# Wait, no: the original list is ["HelloWorld", "App"]. We remove "App" -> becomes ["HelloWorld"] and then
insert "App" at 0 -> ["App", "HelloWorld"]

# However, note: the example says: "first element" is set to "App". So the list becomes [ "App", "HelloWorld" ].

# But what if the list has duplicates? The code removes the first occurrence of the overwrite?
# Actually, the code:
#   context_value.remove(overwrite)    # removes the first occurrence
#   context_value.insert(0, overwrite)
```

```
# So if the list has duplicates, then after removal, one duplicate remains and then we insert at front -> so the
 list will have the overwrite at front and then the rest including the duplicate.

# We should test that.

def test_list_single_choice_with_duplicates():
    context = {"choice_var": ["a", "b", "a"]}
    overwrite = {"choice_var": "a"}
    apply_overwrites_to_context(context, overwrite)
    # It should remove the first occurrence of "a", then insert "a" at front -> becomes ["a", "b", "a"] -> but
after removal: ["b", "a"] and then insert -> ["a", "b", "a"]?
    # Actually:
    #   Original: ['a','b','a']
    #   Remove the first 'a' -> becomes ['b','a']
    #   Insert 'a' at 0 -> ['a','b','a'] -> same as original?
    # But the order is now: the first element is the one we set, and then the rest. But note the duplicate
remains.

    # The expected behavior is to have the chosen value at the front and the rest in the original order?
    # So we expect: ['a','b','a'] -> becomes ['a','b','a']?
    # But note: the removal only removes the first occurrence. So the duplicate remains.
    # The documentation doesn't specify duplicates. But the function is designed to remove one occurrence and
then put one at the front.

    # We'll check:
    assert context["choice_var"] == ["a", "b", "a"]

    # But wait: the function does:
    #   context_value.remove(overwrite)   # removes the first occurrence -> so the list becomes ['b','a']
    #   then insert(0, overwrite) -> ['a','b','a'] -> same as original?

    # Actually, the original had two 'a's. The first element was 'a'. We remove the first element and then put
it again at the front? So the list becomes:
    #   [0] = 'a' (new), [1]='b', [2]='a' (the one that was at index1 originally becomes index2).

    # So the list is the same?

    # But the function does not guarantee uniqueness. It just moves one occurrence to the front.

    # We'll leave it as is.

# We'll also test the multi-choice with duplicates? The multi-choice just replaces the entire list. So
duplicates in the original are lost?
# But the overwrite list might have duplicates? The condition is: set(overwrite).issubset(set(context_value))
# But note: set(context_value) will remove duplicates. So if the original has duplicates, the set is without
duplicates.
# And the overwrite set must be a subset of that set. Then the entire list is replaced by the overwrite list (
which might have duplicates?).

# Example:
#   context_value = ['a','a','b'] -> set is {'a','b'}
#   overwrite = ['a','a'] -> set is {'a'} -> subset? yes. Then the context becomes ['a','a'].

# So we can test:

def test_multi_choice_with_duplicates():
    context = {"multi_choice": ["a", "a", "b"]}
    overwrite = {"multi_choice": ["a", "a"]}
    apply_overwrites_to_context(context, overwrite)
    assert context["multi_choice"] == ["a", "a"]

# Also, if the overwrite has an element that is in the set but the original had duplicates, it's allowed.

# But note: the condition is set(overwrite).issubset(set(context_value)), so duplicates in the overwrite are
allowed as long as the distinct elements are in the context.

# Now, we have covered the main cases.

# We must also test the case when the context_value is a boolean and the overwrite is a string that is in the
yes_choices or no_choices.

# We have done that.

# Let's write the tests accordingly.

# We'll now output the code.

Note: We must import pytest and the function.
```

We'll write the tests in a single block.

One more: test that the function is called recursively and the in_dictionary_variable is set to True for nested dictionaries.

We have a test for that: test_dictionary_recursive_update and test_new_key_in_nested_via_recursion.

We'll also test the case when the context_value is a list and in_dictionary_variable is True (via being in a nested dictionary) and the overwrite is a list? Then it should replace the entire list?

But note: the condition for in_dictionary_variable is checked first, so it will set the list to the overwrite list (or any type) and continue.

We have a test for that: test_list_in_dictionary_variable, but that test uses a string. Let's also test with a list.

```python
def test_list_in_dictionary_variable_with_list_overwrite():
    context = {
        "outer": {
            "list_var": ["a", "b"]
        }
    }
    overwrite = {
        "outer": {
            "list_var": ["c", "d"]
        }
    }
    apply_overwrites_to_context(context, overwrite)
    assert context["outer"]["list_var"] == ["c", "d"]
```

But note: the condition for in_dictionary_variable is true in the recursive call, so the list is replaced without checking the subset? Yes.

We'll add that.

But we already have a test for multi_choice in the top level? So we don't need to test the multi_choice in nested? Because the nested one is replaced without validation.

We'll write the test for the above.

Now, we have a comprehensive set.

We'll output the code.

Note: We must not forget to test the error messages? We are using pytest.raises with match, so we are testing the message.

But note: the error message for the list multi-choice includes the entire list. We use a regex that matches part of the message.

We'll write the code.
</think>
```python
from cookiecutter.generate import apply_overwrites_to_context
import pytest

def test_basic_overwrite_simple_variable():
    """Test basic overwrite of existing top-level variable."""
    context = {"name": "old", "version": 1.0}
    overwrite = {"name": "new"}
    apply_overwrites_to_context(context, overwrite)
    assert context == {"name": "new", "version": 1.0}

def test_new_top_level_variable_ignored():
    """Test new top-level variable is ignored by default."""
    context = {"existing": "value"}
    overwrite = {"new_var": "data"}
    apply_overwrites_to_context(context, overwrite)
    assert "new_var" not in context

def test_new_nested_variable_added():
    """Test new variable added in nested dictionary."""
    context = {"nested": {"key": "value"}}
    overwrite = {"nested": {"new_key": "new_value"}}
    apply_overwrites_to_context(context, overwrite)
    assert context["nested"]["new_key"] == "new_value"

def test_list_single_choice_valid():
```

```
    """Test valid single choice moves value to front."""
    context = {"choices": ["a", "b", "c"]}
    overwrite = {"choices": "b"}
    apply_overwrites_to_context(context, overwrite)
    assert context["choices"] == ["b", "a", "c"]

def test_list_single_choice_invalid():
    """Test invalid single choice raises ValueError."""
    context = {"choices": ["a", "b", "c"]}
    overwrite = {"choices": "d"}
    with pytest.raises(ValueError, match="d provided for choice variable choices"):
        apply_overwrites_to_context(context, overwrite)

def test_list_multi_choice_valid():
    """Test valid multi-choice replaces list."""
    context = {"choices": ["a", "b", "c"]}
    overwrite = {"choices": ["b", "c"]}
    apply_overwrites_to_context(context, overwrite)
    assert context["choices"] == ["b", "c"]

def test_list_multi_choice_invalid():
    """Test invalid multi-choice raises ValueError."""
    context = {"choices": ["a", "b", "c"]}
    overwrite = {"choices": ["a", "d"]}
    with pytest.raises(ValueError, match="multi-choice variable choices"):
        apply_overwrites_to_context(context, overwrite)

def test_list_in_nested_dict_replaced():
    """Test list in nested dict is replaced without validation."""
    context = {"config": {"options": ["x", "y"]}}
    overwrite = {"config": {"options": "new"}}  # String replaces list
    apply_overwrites_to_context(context, overwrite)
    assert context["config"]["options"] == "new"

def test_boolean_valid_conversion():
    """Test valid boolean string conversions."""
    context = {"flag1": False, "flag2": True}
    overwrite = {"flag1": "yes", "flag2": "off"}  # "yes"=True, "off"=False
    apply_overwrites_to_context(context, overwrite)
    assert context["flag1"] is True
    assert context["flag2"] is False

def test_boolean_invalid_conversion():
    """Test invalid boolean string raises ValueError."""
    context = {"flag": True}
    overwrite = {"flag": "maybe"}
    with pytest.raises(ValueError, match="could not be converted to a boolean"):
        apply_overwrites_to_context(context, overwrite)

def test_dict_recursive_update():
    """Test nested dictionary partial update."""
    context = {"settings": {"a": 1, "b": 2}}
    overwrite = {"settings": {"b": 99, "c": 3}}
    apply_overwrites_to_context(context, overwrite)
    assert context["settings"] == {"a": 1, "b": 99, "c": 3}

def test_dict_overwritten_by_non_dict():
    """Test dictionary replaced by non-dict value."""
    context = {"config": {"key": "value"}}
    overwrite = {"config": "simple"}
    apply_overwrites_to_context(context, overwrite)
    assert context["config"] == "simple"

def test_new_key_with_in_dictionary_flag():
    """Test new top-level variable added when flag is True."""
    context = {"existing": "value"}
    overwrite = {"new_var": "data"}
    apply_overwrites_to_context(context, overwrite, in_dictionary_variable=True)
    assert context["new_var"] == "data"

def test_empty_context_and_overwrite():
    """Test empty inputs produce no changes."""
    context = {}
    overwrite = {}
    apply_overwrites_to_context(context, overwrite)
    assert context == {}

def test_example_from_documentation():
    """Test behavior matches provided documentation example."""
```

```
    context = {
        "project_name": ["HelloWorld", "App"],
        "enable_feature": True,
        "settings": {"retry_count": 3}
    }
    overwrite = {
        "project_name": "App",
        "settings": {"retry_count": 5}
    }
    apply_overwrites_to_context(context, overwrite)
    assert context["project_name"] == ["App", "HelloWorld"]
    assert context["settings"] == {"retry_count": 5}

def test_multi_choice_with_duplicates():
    """Test multi-choice with duplicate values in context."""
    context = {"choices": ["a", "a", "b"]}
    overwrite = {"choices": ["a", "b"]}
    apply_overwrites_to_context(context, overwrite)
    assert context["choices"] == ["a", "b"]

def test_boolean_edge_case_empty_string():
    """Test empty string for boolean raises error."""
    context = {"flag": True}
    overwrite = {"flag": ""}
    with pytest.raises(ValueError, match="could not be converted to a boolean"):
        apply_overwrites_to_context(context, overwrite)

def test_boundary_large_nested_structure():
    """Test deep nested structure with multiple levels."""
    context = {"a": {"b": {"c": {"d": 1}}}}
    overwrite = {"a": {"b": {"c": {"d": 2, "e": 3}}}}
    apply_overwrites_to_context(context, overwrite)
    assert context["a"]["b"]["c"] == {"d": 2, "e": 3}
```

# G. Repository Details

| Type | Repository | License | #Instances | #Files | LoC |
|---|---|---|---|---|---|
| | pydata/numexpr | MIT | 3 | 24 | 3.8k |
| | keras-team/keras-preprocessing | GPL-3.0 | 3 | 24 | 4.3k |
| | MichaelGrupp/evo | GPL-3.0 | 3 | 58 | 7.8k |
| | quantumlib/OpenFermion | Apache-2.0 | 1 | 346 | 40.7k |
| | mljar/mljar-supervised | MIT | 4 | 163 | 19.3k |
| | datamllab/rlcard | MIT | 1 | 192 | 12.8k |
| | facebookresearch/fvcore | Apache-2.0 | 1 | 57 | 6.8k |
| | colour-science/colour | BSD-3-Clause | 1 | 654 | 159.9k |
| | pyro-ppl/numpyro | Apache-2.0 | 7 | 179 | 46.6k |
| | ZFTurbo/Weighted-Boxes-Fusion | MIT | 2 | 15 | 2.3k |
| | pollen-robotics/dtw | GPL-3.0 | 1 | 8 | 0.2k |
| | DLR-RM/stable-baselines3 | MIT | 3 | 94 | 14.5k |
| | benedekrozemberczki/karateclub | GPL-3.0 | 1 | 124 | 5.5k |
| | pytorch/captum | BSD-3-Clause | 3 | 216 | 42.2k |
| | uncertainty-toolbox/uncertainty-toolbox | MIT | 22 | 24 | 2.5k |
| | py-why/causal-learn | MIT | 1 | 114 | 14.8k |
| | CodeReclaimers/neat-python | BSD-3-Clause | 1 | 77 | 6.8k |
| | pypose/pypose | Apache-2.0 | 4 | 89 | 7.8k |
| | explosion/sense2vec | MIT | 2 | 19 | 2.0k |
| | docarray/docarray | Apache-2.0 | 3 | 315 | 33.2k |
| | salesforce/Merlion | BSD-3-Clause | 1 | 194 | 22.1k |
| | maxpumperla/hyperas | MIT | 2 | 20 | 1.4k |
| Scientific/Engineering | stanfordnlp/dspy | MIT | 4 | 219 | 25.5k |
| | intelligent-machine-learning/dlrover | GPL-3.0 | 7 | 432 | 66.0k |
| | obspy/obspy | GPL-3.0 | 13 | 609 | 107.2k |
| | automl/SMAC3 | GPL-3.0 | 6 | 209 | 16.8k |
| | qdrant/fastembed | Apache-2.0 | 2 | 76 | 7.3k |
| | Pyomo/pyomo | GPL-3.0 | 3 | 1679 | 363.7k |
| | rushter/MLAlgorithms | MIT | 1 | 69 | 3.4k |
| | tensorly/tensorly | GPL-3.0 | 12 | 159 | 15.8k |
| | topoteretes/cognee | Apache-2.0 | 1 | 843 | 45.4k |
| | keon/algorithms | MIT | 3 | 406 | 13.1k |
| | sympy/sympy | GPL-3.0 | 1 | 1516 | 440.5k |
| | D-Star-AI/dsRAG | MIT | 3 | 84 | 11.4k |
| | microsoft/graphrag | MIT | 2 | 445 | 26.2k |
| | circlemind-ai/fast-graphrag | MIT | 1 | 54 | 6.0k |
| | cornellius-gp/gpytorch | MIT | 5 | 301 | 24.2k |
| | rigetti/pyquil | Apache-2.0 | 4 | 102 | 20.4k |
| | deepcharles/ruptures | BSD-2-Clause | 1 | 48 | 1.9k |
| | python-adaptive/adaptive | BSD-3-Clause | 3 | 41 | 7.2k |
| | pgmpy/pgmpy | MIT | 8 | 169 | 46.1k |
| | fairlearn/fairlearn | MIT | 1 | 163 | 18.8k |
| | mathics/Mathics | GPL-3.0 | 3 | 193 | 50.0k |
| | geomstats/geomstats | MIT | 1 | 519 | 52.7k |
| | sdatkinson/neural-amp-modeler | MIT | 2 | 61 | 7.1k |

| Type | Repository | License | #Instances | #Files | LoC |
|---|---|---|---|---|---|
| Scientific/Engineering | online-ml/river | BSD-3-Clause | 3 | 476 | 31.0k |
| | NanoVNA-Saver/nanovna-saver | Unknown | 6 | 110 | 13.7k |
| | UKPLab/sentence-transformers | Apache-2.0 | 4 | 366 | 39.0k |
| | bayesian-optimization/BayesianOptimization | MIT | 1 | 25 | 3.8k |
| | sdv-dev/CTGAN | GPL-3.0 | 1 | 29 | 2.1k |
| | stumpy-dev/stumpy | GPL-3.0 | 2 | 96 | 23.1k |
| | google-research/text-to-text-transfer-transformer | Apache-2.0 | 2 | 38 | 8.6k |
| | MouseLand/cellpose | BSD-3-Clause | 1 | 54 | 16.8k |
| | google-deepmind/dm-haiku | Apache-2.0 | 1 | 150 | 20.8k |
| | frgfm/torch-cam | Apache-2.0 | 1 | 24 | 1.7k |
| | explosion/thinc | MIT | 3 | 157 | 17.1k |
| | huggingface/trl | Apache-2.0 | 1 | 173 | 38.1k |
| | facebookresearch/fairscale | GPL-3.0 | 1 | 289 | 32.3k |
| | lightly-ai/lightly | MIT | 1 | 738 | 67.1k |
| | jina-ai/finetuner | Apache-2.0 | 1 | 35 | 3.6k |
| | tensorflow/datasets | Apache-2.0 | 1 | 1454 | 113.0k |
| | towhee-io/towhee | Apache-2.0 | 1 | 707 | 41.9k |
| | deepdoctection/deepdoctection | Apache-2.0 | 2 | 229 | 32.3k |
| | WenjieDu/PyPOTS | BSD-3-Clause | 1 | 616 | 43.8k |
| | pytroll/satpy | GPL-3.0 | 1 | 497 | 90.2k |
| | facebookresearch/multimodal | BSD-3-Clause | 2 | 280 | 38.2k |
| | google-deepmind/android_env | Apache-2.0 | 1 | 76 | 11.1k |
| | Lightning-AI/lit-llama | Apache-2.0 | 1 | 48 | 5.4k |
| | explosion/spacy-llm | MIT | 5 | 153 | 12.7k |
| | joblib/joblib | BSD-3-Clause | 2 | 88 | 15.7k |
| | bghira/SimpleTuner | AGPL-3.0 | 1 | 254 | 87.0k |
| | brian-team/brian2 | GPL-3.0 | 4 | 304 | 57.9k |
| | daavoo/pyntcloud | MIT | 3 | 93 | 4.6k |
| | autorope/donkeycar | MIT | 1 | 163 | 22.3k |
| | pytorch/tnt | GPL-3.0 | 6 | 189 | 25.3k |
| | chainer/chainerrl | MIT | 1 | 250 | 25.6k |
| | apple/coremltools | BSD-3-Clause | 1 | 752 | 214.5k |
| Utilities | davidaurelio/hashids-python | MIT | 1 | 3 | 0.4k |
| | python-babel/babel | BSD-3-Clause | 6 | 71 | 12.7k |
| | tobgu/pyrsistent | MIT | 1 | 43 | 6.3k |
| | konradhalas/dacite | MIT | 35 | 29 | 1.9k |
| | skorokithakis/shortuuid | BSD-3-Clause | 1 | 5 | 0.3k |
| | HBNetwork/python-decouple | MIT | 2 | 10 | 0.5k |
| | pimutils/vdirsyncer | GPL-3.0 | 3 | 67 | 7.4k |
| | DeppWang/youdaonote-pull | MIT | 1 | 7 | 1.1k |
| | Jules-WinnfieldX/CyberDropDownloader | GPL-3.0 | 1 | 81 | 8.5k |
| | mewwts/addict | MIT | 7 | 4 | 0.6k |
| | amoffat/sh | MIT | 1 | 4 | 4.2k |
| | prometheus/client_python | Apache-2.0 | 25 | 42 | 6.9k |
| | keleshev/schema | MIT | 23 | 3 | 2.5k |

| Type | Repository | License | #Instances | #Files | LoC |
|---|---|---|---|---|---|
| Utilities | Suor/funcy | BSD-3-Clause | 1 | 34 | 3.0k |
| | pyeve/cerberus | ISC | 4 | 23 | 5.3k |
| | celery/django-celery-beat | GPL-3.0 | 4 | 49 | 4.2k |
| | python-pendulum/pendulum | MIT | 1 | 173 | 18.8k |
| | alecthomas/voluptuous | BSD-3-Clause | 8 | 9 | 3.0k |
| | jd/tenacity | Apache-2.0 | 10 | 20 | 3.2k |
| | kennethreitz/maya | MIT | 4 | 8 | 1.6k |
| | mahmoud/boltons | GPL-3.0 | 2 | 62 | 11.9k |
| | LKI/chinese-calendar | MIT | 1 | 16 | 2.3k |
| | ReactiveX/RxPY | MIT | 12 | 405 | 39.6k |
| | joke2k/django-environ | MIT | 16 | 20 | 2.4k |
| | python-validators/validators | MIT | 9 | 64 | 3.1k |
| | pytransitions/transitions | MIT | 1 | 53 | 10.1k |
| | coursera-dl/coursera-dl | LGPL-3.0 | 1 | 29 | 4.0k |
| | more-itertools/more-itertools | MIT | 3 | 9 | 9.9k |
| | agronholm/apscheduler | MIT | 1 | 66 | 9.6k |
| | arrow-py/arrow | Apache-2.0 | 7 | 20 | 14.8k |
| | aio-libs/aiocache | BSD-3-Clause | 1 | 50 | 4.6k |
| | PyFilesystem/pyfilesystem2 | MIT | 1 | 106 | 13.4k |
| Text Processing | mjpost/sacrebleu | Apache-2.0 | 1 | 42 | 5.2k |
| | vi3k6i5/flashtext | MIT | 3 | 18 | 1.0k |
| | google-research/arxiv-latex-cleaner | Apache-2.0 | 3 | 7 | 1.9k |
| | textstat/textstat | MIT | 22 | 123 | 3.2k |
| | carpedm20/emoji | GPL-3.0 | 7 | 23 | 3.1k |
| | summanlp/textrank | MIT | 4 | 20 | 4.2k |
| | nidhaloff/deep-translator | Apache-2.0 | 2 | 40 | 2.6k |
| | r1chardj0n3s/parse | MIT | 10 | 8 | 1.5k |
| | adbar/trafilatura | Apache-2.0 | 2 | 41 | 21.7k |
| | jsvine/markovify | MIT | 4 | 11 | 0.8k |
| | ssut/py-googletrans | MIT | 2 | 14 | 1.3k |
| | google/textfsm | Apache-2.0 | 1 | 10 | 3.3k |
| | google/budou | Apache-2.0 | 1 | 23 | 1.2k |
| | jaraco/inflect | Unknown | 2 | 17 | 4.8k |
| | pemistahl/lingua-py | Apache-2.0 | 2 | 5 | 0.9k |
| | pyparsing/pyparsing | MIT | 3 | 133 | 22.8k |
| | stanfordnlp/stanza | GPL-3.0 | 1 | 476 | 51.4k |
| | sloria/TextBlob | MIT | 1 | 37 | 4.0k |
| | seatgeek/fuzzywuzzy | GPL-2.0 | 5 | 11 | 1.0k |
| | amperser/proselint | BSD-3-Clause | 1 | 114 | 4.9k |
| | Python-Markdown/markdown | BSD-3-Clause | 5 | 67 | 11.7k |
| Software Development | pew-org/pew | MIT | 1 | 28 | 1.3k |
| | pschanely/CrossHair | GPL-3.0 | 7 | 163 | 32.0k |
| | dropbox/pyannotate | Apache-2.0 | 4 | 25 | 3.9k |
| | google/pinject | Apache-2.0 | 1 | 41 | 3.7k |
| | kapicorp/kapitan | Apache-2.0 | 3 | 86 | 9.3k |

| Type | Repository | License | #Instances | #Files | LoC |
|------|------------|---------|-----------|--------|-----|
| | python-injector/injector | BSD-3-Clause | 11 | 4 | 2.1k |
| | eliben/pyelftools | GPL-3.0 | 5 | 122 | 15.9k |
| | cookiecutter/cookiecutter | BSD-3-Clause | 16 | 88 | 6.2k |
| | nosarthur/gita | MIT | 10 | 11 | 2.2k |
| | PyCQA/isort | MIT | 7 | 97 | 18.7k |
| | SolidCode/SolidPython | Unknown | 1 | 35 | 4.1k |
| | facebookincubator/Bowler | MIT | 5 | 20 | 2.5k |
| | nschloe/tuna | GPL-3.0 | 2 | 12 | 0.6k |
| | dephell/dephell | MIT | 4 | 251 | 15.1k |
| | ekalinin/nodeenv | GPL-3.0 | 2 | 4 | 1.1k |
| | getsentry/sentry-python | MIT | 4 | 398 | 67.4k |
| | sqlalchemy/alembic | MIT | 6 | 101 | 39.5k |
| | olofk/fusesoc | BSD-2-Clause | 3 | 47 | 6.1k |
| | FactoryBoy/factory_boy | MIT | 2 | 54 | 7.4k |
| | pydoit/doit | MIT | 5 | 133 | 11.6k |
| | pre-commit/pre-commit-hooks | MIT | 5 | 69 | 3.6k |
| | PyCQA/pyflakes | MIT | 17 | 21 | 4.0k |
| | pre-commit/pre-commit | MIT | 9 | 129 | 13.6k |
| | facebookresearch/hydra | MIT | 5 | 308 | 32.5k |
| | Shpota/github-activity-generator | Apache-2.0 | 2 | 2 | 0.1k |
| | platformio/platformio-core | Apache-2.0 | 1 | 246 | 29.0k |
| | fastmonkeys/stellar | MIT | 1 | 11 | 0.8k |
| Software Development | zhanyong-wan/dongbei | MIT | 1 | 4 | 2.6k |
| | openapi-generators/openapi-python-client | MIT | 3 | 299 | 19.8k |
| | pythonprofilers/memory_profiler | GPL-3.0 | 2 | 34 | 2.2k |
| | mesonbuild/meson | Apache-2.0 | 4 | 500 | 87.2k |
| | trailofbits/graphtage | LGPL-3.0 | 1 | 51 | 7.3k |
| | koxudaxi/fastapi-code-generator | MIT | 6 | 50 | 1.8k |
| | nbQA-dev/nbQA | MIT | 15 | 60 | 3.8k |
| | breuleux/jurigged | MIT | 3 | 63 | 4.2k |
| | google/yapf | Apache-2.0 | 17 | 69 | 13.5k |
| | bndr/pipreqs | Apache-2.0 | 8 | 14 | 1.1k |
| | nickstenning/honcho | MIT | 11 | 34 | 2.1k |
| | cantools/cantools | MIT | 15 | 79 | 23.7k |
| | dabeaz/curio | GPL-3.0 | 3 | 96 | 10.2k |
| | terryyin/lizard | GPL-3.0 | 7 | 130 | 10.4k |
| | watson-developer-cloud/python-sdk | Apache-2.0 | 5 | 43 | 72.8k |
| | basetenlabs/truss | MIT | 16 | 287 | 36.8k |
| | rubik/radon | MIT | 1 | 29 | 3.8k |
| | guardrails-ai/guardrails | Apache-2.0 | 2 | 327 | 30.6k |
| | Pythagora-io/gpt-pilot | GPL-3.0 | 2 | 130 | 15.5k |
| | SWE-bench/SWE-bench | MIT | 1 | 84 | 13.3k |
| | noamgat/lm-format-enforcer | MIT | 1 | 25 | 2.9k |
| | spulec/freezegun | Apache-2.0 | 11 | 23 | 2.2k |
| | crewAIInc/crewAI-tools | MIT | 7 | 197 | 17.0k |

| Type | Repository | License | #Instances | #Files | LoC |
|---|---|---|---|---|---|
| | aws-ia/taskcat | Apache-2.0 | 1 | 85 | 10.8k |
| | enoch3712/ExtractThinker | Apache-2.0 | 1 | 110 | 12.0k |
| | weaveworks/grafanalib | Apache-2.0 | 15 | 37 | 6.9k |
| | Sceptre/sceptre | GPL-3.0 | 7 | 111 | 15.4k |
| | palantir/python-language-server | MIT | 1 | 62 | 5.1k |
| | nschloe/perfplot | GPL-3.0 | 9 | 7 | 0.6k |
| | qodo-ai/qodo-cover | AGPL-3.0 | 3 | 73 | 10.5k |
| | run-llama/llama_deploy | MIT | 1 | 94 | 5.9k |
| | asottile/pyupgrade | MIT | 14 | 97 | 10.6k |
| | firebase/firebase-admin-python | Apache-2.0 | 5 | 74 | 22.1k |
| | jendrikseipp/vulture | MIT | 6 | 44 | 2.9k |
| | autoscrape-labs/pydoll | MIT | 2 | 105 | 21.4k |
| | BerriAI/litellm | GPL-3.0 | 1 | 1975 | 417.1k |
| | jupyterhub/repo2docker | BSD-3-Clause | 8 | 100 | 8.1k |
| | googleapis/python-genai | Apache-2.0 | 4 | 190 | 55.9k |
| | kronenthaler/mod-pbxproj | MIT | 5 | 77 | 3.9k |
| | langchain-ai/langchain-mcp-adapters | MIT | 1 | 18 | 1.6k |
| | Instagram/MonkeyType | GPL-3.0 | 5 | 34 | 5.9k |
| | Delgan/loguru | MIT | 2 | 168 | 13.5k |
| | gitpython-developers/GitPython | BSD-3-Clause | 2 | 88 | 18.3k |
| | prospector-dev/prospector | GPL-2.0 | 1 | 107 | 5.1k |
| | hhatto/autopep8 | MIT | 7 | 65 | 10.2k |
| Software Development | wemake-services/wemake-python-styleguide | MIT | 2 | 378 | 30.1k |
| | fsspec/filesystem_spec | BSD-3-Clause | 17 | 104 | 24.3k |
| | python-rope/rope | LGPL-3.0 | 7 | 144 | 31.5k |
| | procrastinate-org/procrastinate | MIT | 3 | 169 | 13.9k |
| | itamarst/eliot | Apache-2.0 | 2 | 68 | 8.9k |
| | theskumar/python-dotenv | BSD-3-Clause | 3 | 18 | 2.1k |
| | koxudaxi/datamodel-code-generator | MIT | 20 | 531 | 41.6k |
| | coleifer/huey | MIT | 1 | 52 | 6.1k |
| | aws-powertools/powertools-lambda-python | MIT-0 | 2 | 1091 | 65.3k |
| | nolar/kopf | MIT | 21 | 280 | 29.3k |
| | python-lsp/python-lsp-server | MIT | 2 | 70 | 8.0k |
| | pyocd/pyOCD | Apache-2.0 | 1 | 386 | 61.8k |
| | litestar-org/polyfactory | MIT | 3 | 141 | 11.2k |
| | getnikola/nikola | MIT | 1 | 226 | 23.2k |
| | cpplint/cpplint | GPL-3.0 | 6 | 3 | 8.5k |
| | faif/python-patterns | Unknown | 1 | 56 | 2.0k |
| | kevin1024/vcrpy | MIT | 2 | 69 | 6.4k |
| | conan-io/conan | MIT | 1 | 911 | 98.6k |
| | agronholm/typeguard | GPL-3.0 | 1 | 31 | 6.2k |
| | adrienverge/yamllint | GPL-3.0 | 22 | 68 | 11.5k |
| | PyCQA/flake8-bugbear | MIT | 2 | 70 | 4.4k |
| | eyurtsev/kor | MIT | 4 | 42 | 2.3k |
| | spotify/luigi | Apache-2.0 | 1 | 253 | 38.2k |

| Type | Repository | License | #Instances | #Files | LoC |
|------|-----------|---------|-----------|--------|-----|
| | griptape-ai/griptape | Apache-2.0 | 4 | 1101 | 45.4k |
| | cloudtools/troposphere | BSD-2-Clause | 1 | 575 | 77.4k |
| | ethereum/py-evm | MIT | 1 | 408 | 41.9k |
| | getlogbook/logbook | GPL-3.0 | 1 | 69 | 6.7k |
| | joke2k/faker | MIT | 2 | 737 | 341.5k |
| Software Development | lark-parser/lark | MIT | 5 | 83 | 12.2k |
| | terraform-compliance/cli | MIT | 4 | 54 | 6.7k |
| | BeehiveInnovations/zen-mcp-server | GPL-3.0 | 2 | 194 | 38.8k |
| | pypa/pipenv | MIT | 1 | 643 | 129.0k |
| | Yelp/mrjob | GPL-3.0 | 5 | 218 | 46.4k |
| | getpelican/pelican | AGPL-3.0 | 10 | 50 | 12.8k |
| | SmileyChris/django-countries | MIT | 1 | 31 | 3.5k |
| | novnc/websockify | LGPL-3.0 | 2 | 19 | 3.4k |
| | mar10/wsgidav | MIT | 1 | 54 | 10.1k |
| | strawberry-graphql/strawberry | MIT | 11 | 572 | 58.8k |
| | Kludex/mangum | MIT | 1 | 21 | 3.5k |
| | exentriquesolutions/nip.io | GPL-3.0 | 1 | 5 | 1.1k |
| | Shopify/shopify_python_api | MIT | 12 | 156 | 4.3k |
| | googlemaps/google-maps-services-python | Apache-2.0 | 2 | 31 | 3.0k |
| | alexgolec/tda-api | MIT | 14 | 44 | 12.6k |
| | adamghill/django-unicorn | MIT | 11 | 122 | 8.6k |
| | Pylons/waitress | GPL-3.0 | 21 | 46 | 10.7k |
| | requests-cache/requests-cache | BSD-2-Clause | 8 | 72 | 7.8k |
| | graphql-python/graphene | MIT | 16 | 124 | 8.4k |
| | deschler/django-modeltranslation | BSD-3-Clause | 2 | 35 | 7.4k |
| | zauberzeug/nicegui | MIT | 1 | 701 | 38.0k |
| | TransformerOptimus/SuperAGI | MIT | 7 | 398 | 21.6k |
| Internet | scrapy/scrapyd | BSD-3-Clause | 6 | 56 | 3.3k |
| | koalalorenzo/python-digitalocean | LGPL-3.0 | 15 | 44 | 5.3k |
| | mwmbl/mwmbl | AGPL-3.0 | 6 | 128 | 15.0k |
| | ross/requests-futures | GPL-3.0 | 1 | 5 | 0.4k |
| | DedSecInside/TorBot | GPL-3.0 | 3 | 11 | 0.7k |
| | python-hyper/hyper | MIT | 1 | 46 | 7.2k |
| | psf/requests | Apache-2.0 | 1 | 35 | 7.1k |
| | Pylons/pyramid | GPL-3.0 | 1 | 330 | 45.5k |
| | aiortc/aioquic | BSD-3-Clause | 3 | 58 | 20.0k |
| | MechanicalSoup/MechanicalSoup | MIT | 6 | 20 | 1.9k |
| | django-json-api/django-rest-framework-json-api | BSD-2-Clause | 2 | 77 | 9.0k |
| | michaelhly/solana-py | MIT | 8 | 51 | 6.8k |
| | rthalley/dnspython | GPL-3.0 | 3 | 241 | 36.5k |
| | django/channels | BSD-3-Clause | 2 | 45 | 3.2k |
| | miguelgrinberg/python-socketio | MIT | 3 | 95 | 12.5k |
| | django/asgiref | BSD-3-Clause | 1 | 20 | 2.6k |
| | django/daphne | BSD-3-Clause | 2 | 24 | 2.6k |
| | aws/chalice | Apache-2.0 | 1 | 110 | 37.7k |

| Type | Repository | License | #Instances | #Files | LoC |
|---|---|---|---|---|---|
| Internet | maxmind/GeoIP2-python | Apache-2.0 | 1 | 13 | 2.3k |
| | scrapy/scrapy | BSD-3-Clause | 2 | 390 | 54.5k |
| | graphql-python/gql | MIT | 5 | 117 | 21.3k |
| Security | initstring/linkedin2username | MIT | 2 | 2 | 0.6k |
| | authlib/authlib | BSD-3-Clause | 1 | 289 | 27.2k |
| | casbin/pycasbin | Apache-2.0 | 13 | 85 | 7.8k |
| | a13xp0p0v/kernel-hardening-checker | GPL-3.0 | 1 | 5 | 2.0k |
| | quark-engine/quark-engine | GPL-3.0 | 1 | 76 | 10.3k |
| | obsidianforensics/hindsight | Apache-2.0 | 1 | 21 | 4.8k |
| | GitGuardian/ggshield | MIT | 3 | 250 | 22.7k |
| | salesforce/policy_sentry | MIT | 2 | 77 | 6.3k |
| | CTFd/CTFd | Apache-2.0 | 16 | 304 | 33.6k |
| | oauthlib/oauthlib | BSD-3-Clause | 17 | 137 | 11.7k |
| | jaraco/keyring | Unknown | 2 | 43 | 1.9k |
| | mozilla/bleach | GPL-3.0 | 3 | 53 | 14.6k |
| | linkedin/qark | GPL-3.0 | 5 | 68 | 3.1k |
| | mschwager/fierce | GPL-3.0 | 2 | 4 | 0.9k |
| | django-guardian/django-guardian | GPL-3.0 | 3 | 95 | 8.5k |
| | QIN2DIM/hcaptcha-challenger | GPL-3.0 | 1 | 105 | 8.9k |
| | python-security/pyt | GPL-2.0 | 1 | 248 | 9.2k |
| | yandex/gixy | GPL-3.0 | 2 | 48 | 4.5k |
| | petertodd/python-bitcoinlib | GPL-3.0 | 2 | 52 | 6.9k |
| | OWASP/Nettacker | Apache-2.0 | 1 | 63 | 6.7k |
| | jazzband/djangorestframework-simplejwt | MIT | 1 | 48 | 3.9k |
| | log2timeline/plaso | Apache-2.0 | 3 | 887 | 93.3k |
| | Yelp/detect-secrets | Apache-2.0 | 2 | 158 | 10.5k |
| | OWASP/pytm | GPL-3.0 | 1 | 10 | 3.6k |
| | pallets/itsdangerous | BSD-3-Clause | 3 | 14 | 1.0k |
| | mitre/caldera | Apache-2.0 | 2 | 169 | 16.5k |
| | prowler-cloud/prowler | Apache-2.0 | 4 | 2893 | 360.4k |
| | certtools/intelmq | AGPL-3.0 | 3 | 389 | 36.3k |
| | smicallef/spiderfoot | MIT | 7 | 694 | 57.7k |
| | androguard/androguard | Apache-2.0 | 3 | 66 | 25.2k |
| Database | dbcli/mssql-cli | BSD-3-Clause | 1 | 88 | 9.7k |
| | msiemens/tinydb | MIT | 8 | 20 | 2.1k |
| | datastax/python-driver | Apache-2.0 | 1 | 251 | 51.1k |
| | simonw/sqlite-utils | Apache-2.0 | 11 | 55 | 14.0k |
| | pynamodb/PynamoDB | MIT | 15 | 58 | 14.0k |
| | dbcli/pgcli | BSD-3-Clause | 1 | 66 | 9.4k |
| | duckdb/dbt-duckdb | Apache-2.0 | 18 | 84 | 6.2k |
| | qdrant/qdrant-client | Apache-2.0 | 15 | 178 | 62.9k |
| | Aiven-Open/pghoard | Apache-2.0 | 6 | 65 | 13.6k |
| | pgvector/pgvector-python | MIT | 5 | 72 | 3.8k |
| | jazzband/dj-database-url | BSD-3-Clause | 2 | 3 | 0.8k |
| | graphite-project/whisper | Apache-2.0 | 5 | 19 | 2.7k |

| Type | Repository | License | #Instances | #Files | LoC |
|---|---|---|---|---|---|
| Database | piskvorky/sqlitedict | Apache-2.0 | 2 | 11 | 0.8k |
| | kayak/pypika | Apache-2.0 | 42 | 59 | 11.3k |
| | datafold/data-diff | MIT | 1 | 70 | 13.5k |
| | EnterpriseDB/barman | GPL-3.0 | 9 | 99 | 65.9k |
| | reata/sqllineage | MIT | 1 | 92 | 7.0k |
| | aws/aws-sdk-pandas | Apache-2.0 | 2 | 222 | 43.2k |
| | activeloopai/deeplake | Apache-2.0 | 1 | 55 | 7.3k |
| | art049/odmantic | ISC | 2 | 156 | 8.7k |
| | georgia-tech-db/evadb | Apache-2.0 | 1 | 460 | 35.6k |
| | andialbrecht/sqlparse | BSD-3-Clause | 3 | 34 | 5.2k |
| | jazzband/django-redis | GPL-3.0 | 1 | 45 | 3.8k |
| | kvesteri/sqlalchemy-utils | GPL-3.0 | 1 | 169 | 13.0k |
| | pudo/dataset | MIT | 1 | 10 | 1.4k |
| | RDFLib/rdflib | BSD-3-Clause | 2 | 411 | 57.1k |
| | coleifer/peewee | MIT | 7 | 87 | 37.8k |
| System | Supervisor/supervisor | GPL-3.0 | 4 | 69 | 28.7k |
| | alichtman/shallow-backup | MIT | 2 | 20 | 1.9k |
| | jupyterhub/the-littlest-jupyterhub | BSD-3-Clause | 3 | 44 | 3.9k |
| | containers/podman-compose | GPL-2.0 | 3 | 82 | 11.5k |
| | pyinfra-dev/pyinfra | MIT | 4 | 220 | 21.1k |
| | borgmatic-collective/borgmatic | GPL-3.0 | 15 | 274 | 53.6k |
| | andreafrancia/trash-cli | GPL-2.0 | 3 | 336 | 9.7k |
| | jertel/elastalert2 | Apache-2.0 | 106 | 123 | 33.2k |
| | circus-tent/circus | GPL-3.0 | 2 | 132 | 12.2k |
| | patroni/patroni | MIT | 23 | 162 | 31.8k |
| | svinota/pyroute2 | GPL-3.0 | 2 | 407 | 49.6k |
| | docker/docker-py | Apache-2.0 | 1 | 130 | 21.1k |
| | gpustack/gpustack | Apache-2.0 | 4 | 81 | 87.5k |
| | pexpect/pexpect | GPL-3.0 | 2 | 100 | 8.2k |
| | liquidctl/liquidctl | GPL-3.0 | 9 | 76 | 13.7k |
| | giampaolo/psutil | BSD-3-Clause | 2 | 74 | 19.4k |
| | facebookincubator/submitit | MIT | 1 | 30 | 3.7k |
| | cloud-custodian/cloud-custodian | Apache-2.0 | 3 | 1017 | 182.1k |
| | tomerfiliba/plumbum | MIT | 1 | 78 | 9.9k |
| | canonical/cloud-init | Apache 2.0 | 7 | 571 | 132.4k |
| | overhangio/tutor | AGPL-3.0 | 1 | 81 | 7.1k |
| Communications | martinrusev/imbox | MIT | 12 | 20 | 0.7k |
| | websocket-client/websocket-client | Apache-2.0 | 4 | 28 | 4.2k |
| | hbldh/bleak | MIT | 1 | 67 | 10.6k |
| | wee-slack/wee-slack | MIT | 1 | 92 | 19.6k |
| | ktbyers/netmiko | MIT | 1 | 313 | 18.7k |
| | crossbario/autobahn-python | MIT | 1 | 352 | 40.0k |
| | scrapinghub/slackbot | MIT | 2 | 20 | 1.6k |
| | FreeOpcUa/python-opcua | LGPL-3.0 | 2 | 132 | 216.0k |
| | sendgrid/sendgrid-python | MIT | 1 | 125 | 8.8k |

| Type | Repository | License | #Instances | #Files | LoC |
|---|---|---|---|---|---|
| Communications | taskiq-python/taskiq | MIT | 8 | 139 | 6.8k |
| | bear/python-twitter | Apache-2.0 | 7 | 39 | 6.9k |
| | LonamiWebs/Telethon | MIT | 3 | 141 | 16.1k |
| | jookies/jasmin | Apache 2.0 | 2 | 169 | 29.7k |
| | sshuttle/sshuttle | LGPL-2.1 | 9 | 34 | 6.2k |
| | nats-io/nats.py | Apache-2.0 | 1 | 62 | 14.8k |
| | slackapi/python-slack-sdk | MIT | 11 | 411 | 52.0k |
| | Forethought-Technologies/AutoChain | MIT | 2 | 64 | 3.2k |
| | FreeOpcUa/opcua-asyncio | LGPL-3.0 | 5 | 196 | 317.3k |
| | pinecone-io/canopy | Apache-2.0 | 2 | 136 | 10.1k |
| | celery/kombu | BSD-3-Clause | 2 | 167 | 28.5k |
| | zeromq/pyzmq | BSD-3-Clause | 1 | 172 | 15.4k |
| | letta-ai/letta | Apache-2.0 | 5 | 649 | 100.9k |
| | element-hq/synapse | AGPL-3.0 | 3 | 931 | 255.8k |
| File Formats | lidatong/dataclasses-json | MIT | 14 | 38 | 3.7k |
| | JelteF/PyLaTeX | MIT | 3 | 65 | 4.8k |
| | wireservice/csvkit | MIT | 4 | 44 | 4.9k |
| | alan-turing-institute/CleverCSV | MIT | 2 | 69 | 7.4k |
| | msgpack/msgpack-python | Apache 2.0 | 25 | 24 | 2.1k |
| | lincolnloop/python-qrcode | BSD 3-Clause License | 1 | 35 | 3.0k |
| | pdfminer/pdfminer.six | MIT | 6 | 60 | 20.2k |
| | jcrist/msgspec | BSD-3-Clause | 4 | 56 | 20.3k |
| | scrapy/parsel | BSD-3-Clause | 2 | 14 | 2.0k |
| | pmaupin/pdfrw | GPL-3.0 | 1 | 51 | 3.7k |
| | sripathikrishnan/redis-rdb-tools | MIT | 1 | 21 | 2.8k |
| | globocom/m3u8 | GPL-3.0 | 46 | 20 | 4.6k |
| | martinblech/xmltodict | MIT | 9 | 3 | 1.1k |
| | jsvine/pdfplumber | MIT | 2 | 38 | 6.6k |
| | chezou/tabula-py | MIT | 7 | 13 | 1.3k |
| | mangiucugna/json_repair | MIT | 11 | 26 | 2.0k |
| | landing-ai/agentic-doc | Apache-2.0 | 1 | 13 | 6.0k |
| | oomol-lab/pdf-craft | AGPL-3.0 | 1 | 81 | 6.4k |
| Data Analysis | raphaelvallat/pingouin | GPL-3.0 | 2 | 40 | 7.7k |
| | feature-engine/feature_engine | BSD-3-Clause | 58 | 228 | 26.7k |
| | movingpandas/movingpandas | BSD-3-Clause | 23 | 36 | 7.2k |
| | xflr6/graphviz | MIT | 1 | 75 | 3.7k |
| | petl-developers/petl | MIT | 2 | 168 | 22.4k |
| | pydata/pandas-datareader | GPL-3.0 | 1 | 76 | 6.8k |
| | graphistry/pygraphistry | BSD-3-Clause | 1 | 195 | 37.0k |
| | sinaptik-ai/pandas-ai | GPL-3.0 | 24 | 174 | 13.9k |
| | peerchemist/finta | LGPL-3.0 | 2 | 7 | 2.0k |
| | electricitymaps/electricitymaps-contrib | AGPL-3.0 | 2 | 261 | 34.8k |
| | taynaud/python-louvain | BSD-3-Clause | 1 | 7 | 0.6k |
| | ourownstory/neural_prophet | MIT | 11 | 66 | 14.1k |
| | nteract/papermill | BSD-3-Clause | 3 | 38 | 4.9k |

| Type | Repository | License | #Instances | #Files | LoC |
|------|-----------|---------|-----------:|-------:|----:|
| Data Analysis | microsoft/TaskWeaver | MIT | 13 | 147 | 15.6k |
| | pyecharts/pyecharts | MIT | 1 | 128 | 14.4k |
| | jldbc/pybaseball | MIT | 2 | 130 | 7.6k |
| | intake/intake | BSD-2-Clause | 4 | 99 | 11.0k |
| | unionai-oss/pandera | MIT | 3 | 249 | 47.4k |
| | py-why/dowhy | MIT | 3 | 229 | 30.3k |
| | lmfit/lmfit-py | GPL-3.0 | 4 | 105 | 12.5k |
| | bashtage/arch | GPL-3.0 | 2 | 100 | 28.7k |
| | lux-org/lux | Apache-2.0 | 1 | 91 | 10.0k |
| | rasbt/mlxtend | GPL-3.0 | 16 | 210 | 19.5k |
| | fugue-project/fugue | Apache-2.0 | 1 | 223 | 29.1k |
| | unit8co/darts | Apache-2.0 | 2 | 239 | 81.0k |
| | python-streamz/streamz | BSD-3-Clause | 2 | 38 | 6.7k |
| | holoviz/datashader | BSD-3-Clause | 1 | 106 | 26.9k |
| | vispy/vispy | GPL-3.0 | 4 | 557 | 55.6k |
| | dlt-hub/dlt | Apache-2.0 | 11 | 921 | 147.0k |
| | python-visualization/folium | GPL-3.0 | 1 | 101 | 7.4k |
| Terminals | nbedos/termtosvg | BSD-3-Clause | 2 | 13 | 1.9k |
| | bpython/bpython | GPL-3.0 | 9 | 75 | 11.7k |
| | hauntsaninja/pyp | MIT | 8 | 3 | 1.1k |
| | jorgebastida/awslogs | GPL-3.0 | 1 | 9 | 1.1k |
| | rsalmei/alive-progress | MIT | 2 | 41 | 3.0k |
| | jazzband/Watson | MIT | 1 | 21 | 3.6k |
| | kellyjonbrazil/jc | MIT | 17 | 526 | 44.9k |
| | bee-san/pyWhat | MIT | 2 | 19 | 1.8k |
| | nvbn/thefuck | MIT | 28 | 406 | 10.5k |
| | peterbrittain/asciimatics | Apache-2.0 | 1 | 105 | 16.3k |
| | httpie/http-prompt | MIT | 2 | 33 | 4.4k |
| | jarun/buku | GPL-3.0 | 1 | 27 | 8.9k |
| | mkaz/termgraph | MIT | 14 | 18 | 1.4k |
| | gptme/gptme | MIT | 2 | 164 | 24.8k |
| | laixintao/flameshow | MIT | 3 | 35 | 3.0k |
| | online-judge-tools/oj | MIT | 10 | 32 | 5.4k |
| | python-poetry/cleo | MIT | 2 | 108 | 7.9k |
| | pimutils/khal | MIT | 7 | 57 | 12.6k |
| | LazoVelko/Pokemon-Terminal | GPL-3.0 | 1 | 38 | 1.5k |
| | robusta-dev/holmesgpt | MIT | 3 | 338 | 53.0k |
| | tartley/colorama | BSD-3-Clause | 1 | 23 | 1.4k |
| | tmbo/questionary | MIT | 6 | 59 | 4.1k |
| | docopt/docopt | MIT | 5 | 23 | 1.1k |
| | dylanaraps/pywal | MIT | 7 | 26 | 1.2k |
| | Textualize/rich | MIT | 5 | 188 | 29.3k |
| | insanum/gcalcli | MIT | 4 | 34 | 4.5k |
| Other/Nonlisted Topic | KMKfw/kmk_firmware | GPL-3.0 | 6 | 289 | 20.4k |
| Office/Business | sec-edgar/sec-edgar | Apache-2.0 | 1 | 31 | 3.9k |

| Type | Repository | License | #Instances | #Files | LoC |
|---|---|---|---|---|---|
| Office/Business | mlouielu/twstock | MIT | 2 | 23 | 1.3k |
| | beancount/beancount | GPL-2.0 | 7 | 201 | 24.4k |
| | burnash/gspread | MIT | 2 | 17 | 5.7k |
| | jmcnamara/XlsxWriter | BSD-2-Clause | 9 | 1394 | 55.4k |
| | mintapi/mintapi | MIT | 2 | 21 | 3.1k |
| | almarklein/timetagger | GPL-3.0 | 3 | 28 | 10.5k |
| | brndnmtthws/thetagang | AGPL-3.0 | 2 | 28 | 6.6k |
| | andreroggeri/pynubank | MIT | 5 | 23 | 1.5k |
| Education | cosmicpython/code | GPL-3.0 | 1 | 29 | 1.1k |
| Unknown | fengsp/plan | GPL-3.0 | 1 | 22 | 1.2k |
| | jiaaro/pydub | MIT | 2 | 13 | 3.2k |
| Documentation | scanapi/scanapi | MIT | 1 | 67 | 4.4k |
| | mitmproxy/pdoc | MIT-0 | 6 | 78 | 7.0k |
| Multimedia | Breakthrough/PySceneDetect | BSD-3-Clause | 1 | 49 | 8.3k |
| | quodlibet/mutagen | GPL-2.0 | 3 | 108 | 22.7k |
| | SickChill/sickchill | GPL-3.0 | 1 | 310 | 45.7k |
| | beetbox/beets | MIT | 13 | 212 | 45.7k |
| | mido/mido | MIT | 4 | 74 | 4.0k |
| | ytdl-org/youtube-dl | Unlicense | 1 | 902 | 137.5k |
| | brycedrennan/imaginAIry | MIT | 1 | 337 | 49.5k |
| | pytube/pytube | Unlicense | 4 | 39 | 5.2k |
| | lhotse-speech/lhotse | Apache-2.0 | 1 | 446 | 64.3k |
| | Zulko/moviepy | MIT | 10 | 137 | 11.1k |
| | SYSTRAN/faster-whisper | MIT | 2 | 19 | 3.0k |
| | imageio/imageio | BSD-2-Clause | 3 | 98 | 26.5k |
| | spotDL/spotify-downloader | MIT | 1 | 82 | 10.0k |
| Home Automation | jasonacox/tinytuya | MIT | 1 | 67 | 10.5k |
| | gpiozero/gpiozero | GPL-3.0 | 8 | 148 | 15.0k |
| | SoCo/SoCo | MIT | 4 | 65 | 11.9k |
| | thingsboard/thingsboard-gateway | Apache-2.0 | 1 | 217 | 30.1k |
| Games/Entertainment | niklasf/python-chess | GPL-3.0 | 6 | 23 | 14.3k |
| Desktop Environment | sharkwouter/minigalaxy | GPL-3.0 | 9 | 47 | 6.5k |

