# OpenReview forum: "TestExplora: Benchmarking LLMs for Proactive Bug Discovery via Repository-Level Test Generation"
_ICML.cc/2026/Conference — ICML 2026 regular_

### Official Review · Reviewer_D1uD · 2026-03-01

**Soundness:** 3
**Presentation:** 3
**Significance:** 2
**Originality:** 3
**Overall Recommendation:** 4
**Confidence:** 4

**Summary:**

This paper introduces “proactive bug discovery” as an evaluation objective that has largely been overlooked by prior benchmarks, and elevates the testing task to the repository level. It also provides both Black Box and White Box input settings, together with metrics for evaluating the correctness of generated tests, their bug-revealing capability, and coverage.

**Compliance With Llm Reviewing Policy:**

Affirmed.

**Final Justification:**

My concerns have been adequately addressed.

**Key Questions For Authors:**

1. Can the authors further analyze the reason for the low **F2P** scores? For example, is this mainly due to the difficulty of understanding documentation, the complexity of repository environments, or other factors?

Please also address the other concerns raised in the **Weaknesses** above.

**Limitations:**

No. The paper does not sufficiently discuss its limitations, for example, the issue of **reproducibility cost**.

**Strengths And Weaknesses:**

## Strengths
1. The paper’s task definition is generally clear, and both the overall pipeline and the metric definitions are fairly well specified.
2. The benchmark is relatively complete and comprehensive.
3. The paper adopts an execution-based validation protocol: generated tests are evaluated before and after applying the code patch, and a test is considered valid only if it passes before the patch and fails after the patch. This design ensures that the generated test can indeed detect a real defect.
4. The experimental setup and analysis are fairly detailed.

## Weaknesses
1. The paper lacks information about “token usage, computational cost, and overall reproducibility cost”, which would be important for assessing how difficult the benchmark is to reproduce and how usable it is in practice.
2. Since the benchmark relies heavily on the oracle, and the claim of “proactive bug discovery” is also closely tied to it, the DocAgent-generated documentation requires further validation. The authors provide a 200-example human evaluation and report low patch leakage, which is certainly a positive sign. However, it remains unclear whether the generated documentation introduces stylistic bias, noise, or deviations from the true intent.
3. Although the paper is framed around “proactive bug discovery”, the final validation still depends on the pre/post difference induced by a real patch. This guarantees executability and verifiability, but it also means that the “discovered bugs” in the benchmark are, in essence, still bugs that were already discovered previously, rather than latent defects in an open-ended setting. In other words, this benchmark is a “verifiable approximation” of proactive discovery, not its exact equivalent.

---

> ### Author Rebuttal · Authors · 2026-03-30
>
> # Response to Reviewer D1uD
>
> Thank you for your constructive review. We provide our responses below to address your questions.
>
> ## W1: Reproduction Cost
> To address this, we have quantified the reproducibility costs from two main aspects:
>
> * **API Cost per Call**: The cost of running our benchmark remains highly affordable. We calculate the average cost per call for various models:
>
> |Model|Avg Cost per Call ($)|
> |-|-|
> |Qwen3-Coder-30B|0.001814|
> |Claude-Sonnet-4.5|0.113837|
> |Gemini-2.5-pro|0.062888|
> |GPT-4o|0.010913|
> |GPT-5|0.016930|
> |GPT-5-mini|0.004107|
> |o3-mini|0.018510|
> |o4-mini|0.022469|
> |DeepSeek-R1|0.011348|
> |SWEAgent+GPT-5-mini|0.139827|
> |SWEAgent+o4-mini|0.656453|
>
> * **Environment Cost**: Our evaluation framework does not require expensive GPUs. All our evaluations are conducted on standard virtual machines configured with 16 vCPUs, 64 GB of Memory and ubuntu-latest. **These are standard, widely accessible cloud computing instances that most researchers and practitioners can provision at a very low cost.**
>
> ## W2: Documentation Quality
>
> We thank the reviewer for highlighting the importance of oracle quality. To rigorously address concerns about stylistic bias, noise, and intent deviations, we expand on our validation. Specifically, we use GPT-5 to comprehensively audit all 1,552 PRs and conduct a manual review of 300 randomly sampled cases:
>
> |Checker|Total Instance|Missing Functionality|Stylistic Bias|Information Noise|
> |-|-|-|-|-|
> |GPT-5|1,552|7|4|2|
> |Human|300|0|0|0|
>
> Although the documentation is generated by o3-mini, models from the same ecosystem (e.g., GPT-4o) do not exhibit an advantage over Claude-Sonnet-4.5. This indicates that **TestExplora assesses objective testing capabilities rather than stylistic alignment.**
>
>
> ## W3: Discovered Bugs
>
> We appreciate the reviewer's insightful characterization. We agree that regarding dataset construction, our benchmark acts as a verifiable approximation, as relying on historical patches is currently the only scalable way to provide reliable, automated ground truth.
>
> **From a testing methodology and model perspective, this remains a genuine proactive discovery process.** As in real-world software engineering, proactive testing is rarely a blind, open-ended search across an entire repository; it is practically anchored to specific features, public APIs, or modules (just as human developers target specific functions to test).
>
> **TestExplora faithfully replicates how human developers proactively hunt for latent bugs.** This is ensured by providing specific test entry inferences (anchors), strictly withholding all defect-related signals, and relying solely on documentation-derived intent
>
>
> **Our continuous data collection framework guarantees evaluation timeliness and out-of-distribution validity.**
> By constantly integrating newly merged PRs, we ensure that models are always evaluated on fresh defects that are truly unknown to their training memory.
>
>
> ## W4: Reason for Low F2P
>
> We conduct several deeper analyses. To isolate the models' test-writing capabilities from bug-discovery capabilities, we run an ablation experiment where SWEAgent is explicitly provided with the bug fix (patch leakage):
>
> | Model|Method|HP|F2P|EC|CFG|Num.|
> |-|-|-|-|-|-|-|
> |gpt-5-mini|SWEAgent|93.81|17.27|65.45|47.43|23.09|
> ||SWEAgent (patch leakage)|98.26|73.94|59.13|53.73|34.33|
>
> When the model is hinted at the exact location of the bug, the F2P score jumps dramatically from 17.27 to 73.94. **This confirms that while models can write reproduction tests, they lack the autonomous reasoning to locate latent bugs.**
>
> We also analyze the tool-use distribution of the agents to see how they attempt to understand the codebase:
>
> |Tool|view|submit|pytest|python|grep -R|
> |-|-|-|-|-|-|
> | GPT-5-mini |56.0%|7.3%|2.4%|2.0%|2.0%|
> | o4-mini |48.4%|8.1% |<0.2%|1.2%|11.5%|
>
> The data shows that the stronger model (GPT-5-mini) relies more on active execution feedback (pytest, python), whereas the weaker model (o4-mini) relies heavily on static navigation (grep -R). **This indicates that execution-based feedback loops are critical for mimicking human-like debugging.**
>
> In addition, we also compute the F2P pass@k metrics for SWEAgent in Appendix E.2:
>
> |Model|pass@1|pass@2|pass@3|pass@4|pass@5|
> |-|-|-|-|-|-|
> |SWEAgent+o4-mini|9.4|14.8|17.0|20.0|21.2|
> |SWEAgent+GPT-5-mini|17.3|23.6|26.4|28.2|29.7|
>
> These findings suggest that **advancing proactive bug discovery requires a paradigm shift from static code navigation toward execution-driven exploration strategies** — a direction our benchmark is uniquely positioned to evaluate.

---

> > ### Author Rebuttal · Reviewer_D1uD · 2026-04-03
> >
> > I appreciate the response and effort in addressing the review concerns. My concerns have been solved, thus I keep my positive scores.

---

> > > ### Author Response · Authors · 2026-04-03
> > >
> > > Dear Reviewer D1uD,
> > >
> > > Thank you for your constructive review and recognization of our work. Your feedback on reproducibility cost, documentation quality, and the nature of proactive bug discovery helped us strengthen the paper significantly. We're glad our responses addressed your concerns, and we truly appreciate your time and effort throughout the review process.
> > >
> > > Best regards,
> > >
> > > The Authors

---

### Official Review · Reviewer_Qa9D · 2026-03-10

**Soundness:** 2
**Presentation:** 2
**Significance:** 3
**Originality:** 3
**Overall Recommendation:** 4
**Confidence:** 4

**Summary:**

The paper introduces TestExplora, a benchmark framework that evaluates LLMs’ ability to generate tests for discovering proactive bugs in repositories. The framework first craws and filters valid PRs from candidate repos, collecting the golden patch and tests, then builds virtual machines for execution. Instead of giving the PR, which may serve as ground truth and constrain the model, the framework feeds the LLMs with code implementation, documents, and dependencies (doc only for blackbox setting), and uses the golden tests and patches as ground truth to check if generated tests can fail buggy code while passing the patched version. Their experiments show that existing models perform moderately on this task, while using an agentic framework can improve the quality of unit test generation.

**Compliance With Llm Reviewing Policy:**

Affirmed.

**Final Justification:**

My final recommendation for this paper is 4: Weak Accept. In general, I think the paper has reasonable significance, originality, and soundness as mentioned in my initial review.
In the rebuttal phase, the authors conducted additional experiments to address the soundness concerns I raised in the initial review: one-test constraint test generation, analysis of data before and after 2023, and the influence of document quality on the generated tests. In general, my main concerns have been addressed by the author's efforts during the rebuttal session, and I’m fine with increasing the recommendation score from 3 to 4.

**Key Questions For Authors:**

- 1. In the model inputs, I wonder if the documents of dependencies are also provided, and if not, I wonder how adding those documents would influence the evaluation.
- 2. Could you please justify the concern about the number of tests generated in W1?
- 3. Could you please justify your analysis as mentioned in W2?
- 4. Could you please justify the concern as mentioned in W4 and W5?

I am fine to raise the score if my concerns and questions are properly addressed.

**Limitations:**

The paper doesn’t have a section summarizing its limitations, but several concerns have been mentioned in the weaknesses.

**Strengths And Weaknesses:**

## Strengths

### Significance & Originality & Soundness
The motivation of this paper makes sense. The insight of using PRs during the test generation prompt may work as ground truth and influence the model performance, making sense, and it is worth evaluating the capabilities of LLMs in test generation without any PR information. Also, the method proposed by the paper is generally sound.


## Weaknesses

### Presentation
- A Typo in Figure 2: “Molde” should be “Model”.
- Duplication of SWE-Bench references (Line 459 and Line 464).
- On line 127, “Moreover, unlike SWTBench, which is derived from SWEBench (Yang et al., 2024b),” the citation of SWE-Bench is assigned SWE-Agent.
- Organization of Section 4.1: Currently, it is a bit mixed and lacks highlights on each of the focusing points. Please consider better organizing this section (e.g., using subsections to answer each RQ separately).

### Soundness
I have several concerns regarding the experimental settings and analysis.
- 1. Number of tests generated. It seems that the paper didn’t require the LLMs to generate a specific number of tests, which, in terms of benchmarking the ability, is a bit unnatural to me. Generating a bunch of tests at once without specifying number requirements may influence test quality and introduce confounds. I believe it would be helpful if the paper could also evaluate the LLMs by asking them to generate one test for each request.
- 2. On around line 353, you mentioned “existing models have been trained on repositories related to SWEBench, which leads to their relatively stronger performance on pre-2023 data.” and “For GPT-4o, its performance remains relatively poor, which is because the selected model (2024-05-13) is not specifically trained on SWE-type datasets.” However, you also mentioned that your repo collections are mutually exclusive from SWEBench to mitigate data leakage, which seems to conflict with your analysis around line 353.
- 3. The ablation study misses the influence of dependencies in the whitebox setting.

- 4. Though discussed in the ablation study, I am still concerned about the influence of the documentation in the test generation prompt under the paper’s scenario: if we provide high-quality documentation to the LLM without indicating that the program-under-test should be buggy, would the LLM tend to be conservative? That’s to say, if the LLM precisely describes the functionality of the program-under-test, would it make the LLM tend to verify those documented specifications?

- 5. Follow-up to W4: I’m fundamentally concerned about the design of using LLM-generated documents as a reference oracle. The design means that your evaluation results are highly dependent on the generated documents, which is an inherent bias in your benchmark. I am not sure to what extent you could address this bias through ablation.

---

> ### Author Rebuttal · Authors · 2026-03-30
>
> # Response to Reviewer Qa9D
> Thank you for pointing out presentation and limitations; we promise to make revisions. Below, we address your other concerns.
>
> ## W1&Q2: Number of Tests Generated
>
> To address your concern, we conduct an additional experiment evaluating the LLMs constrained to generate exactly one test:
>
> * **Quantitative results on test constraints**:
>
> We evaluate the models under the strict one-test constraint:
>
> |Model|GPT-5-mini|o4-mini|o3-mini|Gemini-2.5-pro|DeepSeek-R1|Claude-Sonnet-4.5|
> |-|-|-|-|-|-|-|
> |Base F2P|11.84|6.45|7.07|5.08|3.03| 16.06|
> |One Test F2P|12.42|6.67|6.97|4.55|2.42|15.76|
>
> **This indicates that artificially capping the test count does not improve test quality.**
>
> * **Qualitative Case Study**:
>
> Qualitatively, we find that LLMs embed multiple distinct assertions into a single test function, circumventing the one-test constraint.
>
> ```python
> from sacrebleu.compat import corpus_bleu
> import pytest
>
> def test_corpus_bleu_basic_case_and_smoothing_validation():
>     ...
>     # 1) Perfect match (list-of-segments form). Expect BLEU == 100.0
>     ...
>     assert pytest.approx(100.0, rel=1e-6) == res.score
>
>     # 2) Lowercasing behavior with single-string inputs:
>     # sys_stream as a single string and ref_streams as a single string are supported.
>     # When lowercase=True, "the" vs "The" should match -> 100.0
>     ...
>     assert pytest.approx(100.0, rel=1e-6) == res_lower.score
>
>     # 3) Invalid smoothing method should be rejected (implementation must validate allowed methods)
>     ...
>     with pytest.raises(ValueError):
>         ...
> ```
>
> Additionally, **this setting aligns with established  benchmarks like SWTBench**, which evaluate the entire generated test without fixed numerical constraints.
>
>
> ## W2&Q3: Data Leakage
>
> We thank the reviewer for catching this imprecise phrasing, which we will correct in the revision. We clarify that mutually exclusive prevents direct leakage (memorizing specific PRs/patches). **The temporal analysis targets a different phenomenon: indirect leakage from training on similar code patterns and repositories.** These are two distinct issues—repo exclusion addresses the former, while our temporal decay analysis quantifies the latter. This distinction actually strengthens our contribution: even after removing direct overlap, indirect exposure still inflates performance on older data, validating the need for a continuously updated benchmark.
>
> ## W3&Q4: Dependencies Analysis
>
> We would like to gently clarify that this analysis is indeed included in the paper. Specifically, Figure 6 illustrates how model performance varies depending on the distribution of dependencies provided in the White Box setting, specifically breaking them down into "only inherits" and "only invokes" conditions. Furthermore, the text on page 8 (L393) discusses this ablation directly, concluding that "models exhibit distinct dependency preferences".
>
> ## W4-5&Q4: Documentation
>
> We address this in two parts:
> * **About Conservative**:
> 	* We deliberately avoid explicitly instructing the LLM that the program is buggy, because providing **such a strong hint would downgrade our proactive discovery paradigm back to the reactive reproduction paradigm.**
> 	* To ensure the reliability of this approach, we verify the documentation quality through both human and automated audits. We manually review 300 additionally sampled cases and use GPT-5 to audit the full set of 1,552 PRs. These evaluations confirm the documentation accurately reflects the intended functionality. **The error rates are minimal**:
>
>
> | Checker | Total Instance | Missing Functionality  |
> | ------- | -------------- | ----------------- |
> | GPT-5   | 1,552          | 7                |
> | Human   | 300            | 0                 |
>
> Therefore, the documentation describes the intended behavior. If a model acts as a diligent developer and rigorously verifies all documented specifications against the code, the test will inherently fail, thereby exposing the hidden defect.
>
> * **About Bias**: We also conduct an experiment to determine whether there is any bias. Due to the word limit, please refer to our response to Reviewer uuKR, W1: Potential Bias of DocAgent, for a more detailed discussion.
>
>
> ## Q1: Documents for Dependencies
>
> Thanks for your advice, we add an experiment on GPT-5-mini with documents for dependencies:
>
> | Model| Method| HP | F2P   | EC | CFG | Num.  |
> |-|-|-|-|-|-|-|
> | gpt-5-mini | SWEAgent   | 93.81  | 17.27 | 65.45   | 47.43    | 23.09 |
> |  | SWEAgent w/ dep. doc.  | 96.71  | 19.70 | 67.50       | 47.82   | 20.51 |
> | gpt-5-mini | Base Model   | 80.25  |  11.84 | 65.78       | 44.37    | 7.78 |
> |  | Base Model w/ dep. doc.  | 75.93 | 9.29 | 65.30      | 44.24  | 13.54 |
>
> This indicates that while SWEAgent effectively explores and utilizes documents for dependencies, passively feeding them to the base model may cause context overload, **thereby validating the necessity of the agent's exploration mechanism.**

---

> > ### Author Rebuttal · Reviewer_Qa9D · 2026-03-31
> >
> > Thanks to the authors for their reply. I still have the following concerns:
> >
> > ## W2&Q3: Data Leakage
> > I’m still not convinced, as this is not a trivial claim to make. Saying existing models train on SWE-Bench-related code requires more references or experiments to prove it. Your current setting is too simple and doesn't control for confounds. For example, you should prove that your data before 2023 shows higher similarity to SWE-Bench than that after 2023 to support your claim.
> >
> > ## W4-5&Q4: Documentation
> > I feel like my concern about conservative test generation is not fully addressed. I was not questioning whether you should indicate bug existence or whether the documentation is accurate. I’m wondering whether your document-as-oracles design will inherently prevent test generation from uncovering new bugs by: *“That’s to say, if the LLM precisely describes the functionality of the program-under-test, would it make the LLM tend to verify those documented specifications?”* For instance, is it possible that the more precise your documents are, the more conservative your test generation is?

---

> > > ### Author Response · Authors · 2026-04-01
> > >
> > > # Response to Reviewer Qa9D's Follow-up
> > > We sincerely appreciate your prompt feedback. Below is our clarification.
> > >
> > >
> > > ## W2&Q3: Data Leakage
> > > We agree that this claim requires stronger evidence, and we conduct additional experiment:
> > >
> > > * **Patch Similarity Analysis**: We first measure the TF-IDF cosine similarity between the PR patches in TestExplora and those in SWEBench. The centroid cosine similarities are:
> > >
> > > |Subset|Similarity to SWEBench|
> > > |-|-|
> > > |Pre-2023|0.7793|
> > > |Post-2023|0.7230|
> > >
> > > This confirms that pre-2023 data in TestExplora is stylistically more similar to SWE-Bench than post-2023 data.
> > >
> > > * **Black Box vs. White Box Temporal Analysis**: To further disentangle memorization, we extend our temporal analysis by comparing the Black Box setting (no source code provided) against the White Box setting (source code and dependencies provided):
> > >
> > > |Model|Context|GPT-4o|o3-mini|o4-mini|Gemini-2.5-pro|Qwen3-Coder-30B|GPT-5-mini|
> > > |-|-|-|-|-|-|-|-|
> > > |Pre-2023|Black Box|1.74|4.63|3.14|2.62|2.29|5.93|
> > > ||White Box|1.94|4.38|6.39|2.64|1.29|7.88|
> > > ||$\Delta$|+0.20|−0.25|+3.25|+0.02|−1.00|+1.95|
> > > |Post-2023|Black Box|0.57|2.19|1.13|0.57|1.41|3.02|
> > > ||White Box|1.29|3.72|4.67|2.46|1.66|6.58|
> > > ||$\Delta$|+0.72|+1.53|+3.54|+1.89|+0.25|+3.56|
> > >
> > > This experiment controls for the confound as follows:
> > > * **In the Black Box setting**, where models only have the information of the signature of tested code, the performance gap between pre-2023 and post-2023 is substantial (e.g., o3-mini: 4.63 vs. 2.19). This is consistent with models having greater familiarity with older code patterns.
> > > * **In the White Box setting**, where models are given the relevant source code and dependencies, the pre/post-2023 gap narrows considerably (e.g., o3-mini: 4.38 vs. 3.72). When models have explicit access to the code under test, their reliance on memorized patterns diminishes.
> > > * **The White Box improvement ($\Delta$) is consistently larger for post-2023 data**. For instance, GPT-5-mini gains +1.95 on pre-2023 but +3.56 on post-2023; o3-mini gains −0.25 on pre-2023 but +1.53 on post-2023. This asymmetry indicates that for pre-2023 data, models already gain enough from training to perform reasonably even without source code, whereas for post-2023 data, providing source code is far more critical.
> > >
> > > **Taken together, our further validates the need for a continuously updated benchmark like TestExplora.**
> > >
> > > ## W4-5&Q4: Documentation
> > >
> > > Thank you for the insightful follow-up. We provide two pieces of empirical evidence:
> > >
> > > * **DocAgent Documentation vs. Human-Written Documentation**: We first compare the average word count of the two documentation sources: human-written documentation average ~1,092 words, while DocAgent-generated documentation averages ~3,490 words. As shown in Table 5 of our paper, switching to the more detailed DocAgent documentation consistently improves both F2P and EC across all models (e.g., GPT-5-mini: EC +10.52; o3-mini: EC +6.20). The improvement in EC is particularly telling, as it directly measures the number of code entry points exercised by the generated tests. If more detailed documentation made models more conservative, we would expect EC to decrease or remain flat. This demonstrates that richer documentation provides models with more signals about program behavior, enabling them to generate tests that cover more edge cases.
> > >
> > > * **Positive Correlation Between Documentation Length and Test Generation Scale**:
> > >
> > > We further analyze the relationship between DocAgent documentation word count and the number of lines in model-generated tests:
> > >
> > > |Word of Doc|GPT-4o|o3-mini|o4-mini|Gemini-2.5-pro|Qwen3-Coder-30B|GPT-5-mini|
> > > |-|-|-|-|-|-|-|
> > > |0~2k|90.96|176.47|209.55|290.66|333.94|243.04|
> > > |2~4k|140.97|298.16|252.43|345.57|421.21|280.50|
> > > |4~6k|173.27|384.67|302.99|394.75|527.91|345.88|
> > > |6~8k|265.67|412.67|314.64|411.82|532.97|361.51|
> > > |8~10k|236.00|462.50|470.49|597.44|766.65|498.75|
> > > |10k+|285.34|491.00|486.23|591.63|813.26|497.70|
> > >
> > > Across all models, documentation length exhibits a clear positive correlation with test code length. More detailed documentation prompts models to generate richer and more extensive tests, rather than more conservative output.
> > >
> > > Additionally, we apologize for the confusion in our previous response. We clarify the role of documentation in TestExplora's design. Documentation describes the post-patch (correct) behavior, while the code under test is the pre-patch (buggy) version. In this setting, a test that faithfully encodes documented specifications will naturally fail on buggy code. Because the bug deviates from the documented intent. This distinguishes TestExplora from benchmarks like SWTBench, where models receive explicit bug signals (e.g., issue reports). We acknowledge that TestExplora evaluates specification-based testing, a well-established paradigm in software engineering, yet current models' F2P below 18% demonstrates that even this capability remains far from solved.

---

### Official Review · Reviewer_nBG4 · 2026-03-11

**Soundness:** 3
**Presentation:** 3
**Significance:** 3
**Originality:** 3
**Overall Recommendation:** 3
**Confidence:** 4

**Summary:**

This paper introduces TestExplora, a repository-level benchmark for LLM-based test generation aimed at evaluating proactive bug discovery (rather than issue-conditioned reproduction). The benchmark is built from real GitHub PRs, uses a filtering pipeline (including reachability-based filtering), and evaluates generated tests with an F2P (fail-to-pass) criterion across pre-fix and post-fix versions. The paper also includes an execution/evaluation infrastructure and broad baseline experiments (including agentic systems).
I find the problem setting meaningful and the benchmark/infrastructure effort substantial. However, I am not yet convinced the paper sufficiently validates the central construct it claims to measure. The empirical section is broad, but many conclusions are stronger than the evidence supports.

**Compliance With Llm Reviewing Policy:**

Affirmed.

**Key Questions For Authors:**

1.	Please precisely define the reachability notion used (static/dynamic/hybrid, function-level/line-level, how explicit call paths are constructed).
2.	Did you estimate false positives / false negatives of the reachability filter (e.g., via manual audit)?
3.	Can you report how filtering changes the task distribution from the raw PR candidate pool to the final benchmark?
4.	For agentic vs. non-agentic results, what resources are matched (token budget, execution budget, tool access, retries)?
5.	Can you add benchmark-construction ablations (e.g., removing reachability filtering / consistency filtering)?
6.	Can you provide uncertainty/variance estimates for key comparisons?

**Limitations:**

Yes

**Strengths And Weaknesses:**

Strengths
- Important problem framing: evaluating proactive bug discovery is a real and underexplored need.
- Substantial benchmark/infrastructure contribution: repository setup, execution harness, and scalable evaluation pipeline appear technically valuable.
- Broad empirical coverage: multiple models/settings, plus useful analyses (e.g., agentic systems, error analysis).
- Practical value: even with caveats, this benchmark may be useful to the community.

Weaknesses
- Construct validity is not sufficiently established: the benchmark relies on documentation (often DocAgent-generated) as an oracle and F2P as a proxy for proactive discovery, but the paper does not fully justify that this combination cleanly measures the claimed capability. Moreover, the generated doc by DocAgent is very detailed (as shown in Appendix B), such extensive documentation is rarely found in real-world practice. Even if test generation tools achieve strong results on this dataset, what are the realistic use cases? This raises concerns about whether the constructed dataset truly reflects real-world challenges.
- Reachability filtering is over-interpreted: the filtering step seems useful as denoising, but structural/call-path reachability does not imply semantic bug triggerability. The paper treats this step too much like a validity guarantee.
- Filtering bias is under-analyzed: the pipeline likely shifts the task distribution (toward more localized/unit-test-friendly cases), but the paper does not clearly quantify raw-pool vs. final-benchmark distribution shift.
- Experimental conclusions are sometimes too strong:
    - agentic vs. non-agentic comparisons are not cleanly controlled (multiple variables change simultaneously),
    - time-bucket analyses are suggestive but not causal evidence for contamination-related claims,
- Missing key ablations: benchmark-construction choices (especially reachability filtering / consistency filtering) are central but not directly validated via ablation.

---

> ### Author Rebuttal · Authors · 2026-03-30
>
> # Response to Reviewer nBG4
> Below, we address your concerns.
>
> ## W1: Construction Validity
> We clarify from three aspect:
> * **Our benchmark follows the TDD paradigm:** detailed specifications are the starting point. Uniform documentation isolates a model's core test-generation capability from the confounding variable of documentation quality variance across repositories.
> * **F2P is well-motivated for proactive bug discovery:** an F2P test fails on the buggy version and passes on the patched version, meaning it precisely triggers the latent bug without prior knowledge of the patch.
> * **Empirical validation of construct validity via human-written documentation:** the Spearman rank correlation on the aggregate model ranking across HP, F2P, EC, and CFG is ρ = 0.90 (p = 0.015).
> * **Realistic two-stage workflow:** Due to the word limit, please see our response to Reviewer uuKR (W1) for details.
>
> ## W2-3,5&Q1-3,5: Filtering Method
>
> We first clarify the **necessity of filtering**:
> * **Real-world pull requests are notoriously messy.** Prior studies demonstrate that at least 15% of all bug fixes consist of multiple tangled changes [1]. Therefore, filtering composite commit is a critical necessity for dataset quality [2, 3, 4].
> * **Our reachability filter is strictly designed to mitigate this noise.** Crucially, the semantic triggerability of the bug is enforced by our dynamic Reference Oracles. Structural reachability acts as a pre-filter; our execution mechanism validates semantic viability by running the test to confirm it dynamically exposes the bug. Meanwhile, consistency filtering ensures that the intention remains unchanged.
> * **Removing reachability filtering would reintroduce ~43% infeasible instances (validated on a 100-sample audit)**. Conducting an end-to-end ablation on such a noisy pool would yield misleading conclusions, as model performance differences would be confounded by the proportion of inherently unsolvable tasks. We therefore report the filtering impact via distribution and manual audit.
>
> Secondly, we examine the validity of filtering methods:
>
> * **Reachability Notion**: Our reachability filtering operates statically. We leverage the established parsing methodology from OpenHands to construct a static call graph of the repository. **A Pull Request is retained only if there is an explicit call path from the test functions in the test patch to the repaired functions in the code patch.**
> * **Accuracy Estimation**: Because our pipeline relies on strict static explicit call paths, our reachability filter prioritizes precision to prevent unresolvable tasks from entering the benchmark. To validate this, we conduct a manual audit on a random sample of 100 instances and **there is no incorrectly retained sample**.
> * **Distribution Estimation**: As shown in the table below, the proportional representation of repository categories remains remarkably stable, **confirming no significant domain-level bias is introduced**:
>
> |Category|Original Percent|Reachability Filtering|Consistency Filtering|
> |-|-|-|-|
> |Software Development|22.0%|22.8%|21.1%|
> |Scientific/Engineering|16.0%|15.1%|16.8%|
> |Internet|7.7%|8.3%|6.9%|
> |Utilities|6.6%|5.8%|7.2%|
> |Security|6.2%|6.5%|5.4%|
> |Data Analysis|6.2%|5.7%|7.1%|
> |Others|35.3%|35.8%|35.5%|
>
> ## W4&Q4: Experimental Conclusion
> * We agree that the agentic vs. non-agentic evaluation is a system-level comparison **demonstrating overall workflow potential**, rather than a strictly controlled ablation study matched for budgets. We will change our claims and explicitly detail the difference in the revision.
> * Regarding the time-bucket analysis, we concur that the trend is suggestive than definitive as causal evidence of data leakage. However, we believe it underscores the necessity of TestExplora's continuous framework.
>
> ## Q6: Variance for Comparisons
>
> We report the variance of White Box setting due to space limit:
>
> |Model|HP (var)|F2P (var)|EC (var)|CFG (var)|Num. (var)|
> |-|-|-|-|-|-|
> |Qwen3-Ccoder-30B|21.74|0.08|2.84|0.47|0.13|
> |o3-mini|10.45|0.75|4.93|0.07|0.04|
> |o4-mini|5.64|0.28|0.02|0.55|0.01|
> |Gemini 2.5-Pro|2.05|0.11|15.75|0.32|0.13|
> |GPT-4o|13.15|0.61|6.19|0.18|0.05|
> |GPT-5 mini|3.30|1.20|2.29|0.84|0.01|
>
> The HP of some models might be slightly higher. Therefore, we list the results from the three runs as follows:
>
> **o3-mini**:
>   - Run 1: 67.01
>   - Run 2: 73.20
>   - Run 3: 74.38
>
> **GPT-4o**:
>   - Run 1: 52.81
>   - Run 2: 60.10
>   - Run 3: 60.85
>
> **Qwen3-Ccoder**:
>   - Run 1: 54.35
>   - Run 2: 63.52
>   - Run 3: 64.83
>
> **The low variance across F2P, EC, and CFG confirms our pipeline's stability and reproducibility.**
>
> $Ref$
>
> [1] The Impact of Tangled Code Changes, MSR13.
>
> [2] Detecting and Untangling Composite Commits via Attributed Graph Modeling, JCST25.
>
> [3] LLM-Based Detection of Tangled Code Changes for Higher-Quality Method-Level Bug Datasets, Arxiv25.
>
> [4] Atomizer: An LLM-based Collaborative Multi-Agent Framework for Intent-Driven Commit Untangling, Arxiv26.

---

> > ### Author Rebuttal · Reviewer_nBG4 · 2026-04-02
> >
> > I remain concerned about the practical usability of the proposed approach. Even if an LLM achieves strong performance on this benchmark, that does not necessarily indicate its capability to detect real-world bugs. In practice, reliable documentation is often unavailable.
> >
> > The authors claim a realistic two-stage workflow, but I see a fundamental paradox in it. Specifically, in practice, suppose the function under test contains a bug. The tool would first infer documentation from the buggy code, and then use the inferred (buggy) documentation to generate tests aimed at detecting the bug. This creates a circular dependency that undermines the claimed realism.

---

> > > ### Author Response · Authors · 2026-04-02
> > >
> > > # Response to Reviewer nBG4's Follow-up
> > >
> > > We thank the reviewer for the continued engagement. We address the two concerns below.
> > >
> > > * **Clarifying the benchmark construction (no circular dependency)**: We appreciate the reviewer raising this point, and we believe the concern may stem from a misunderstanding of our pipeline. **In our actual pipeline, DocAgent generates documentation from the head (fixed) commit, not the buggy version (L410–411).** It explores the fixed codebase to infer intended functionality, and it does not access diff patches or error messages, as validated in Section 5. The evaluation then checks whether LLM-generated tests, based on this correct specification, can detect deviations in the buggy version. There is no circular dependency. We guarantee that we will re-emphasized the relevant sections in the main text.
> > >
> > > * **Why this design reflects real-world practice**: While the concern about documentation availability is understandable, we note that the landscape has shifted considerably:
> > > 	*  In modern AI-assisted development, high-quality textual descriptions of intended behavior are increasingly common and, in many workflows, required. On the specification side, spec-driven development has become a mainstream methodology in 2025, where tools such as AWS Kiro [1] and GitHub's Spec Kit [2] require developers to produce structured requirements and design documents before any code is generated. On the documentation side, AI coding agents such as Anthropic's Claude Code maintain persistent project-level documentation (CLAUDE.md) that is loaded into every session [3]. Whether expressed as formal specifications or project-level documentation, such descriptions are no longer optional but foundational to modern development practice.
> > > 	* Our benchmark directly evaluates a natural and well-defined step in this workflow: given a correct specification, can an LLM write tests that expose bugs in a faulty implementation? This mirrors the logic of Test-Driven Development, where specs are written before or alongside code and serve as the ground truth for correctness. By generating documentation from the head (fixed) commit, we simulate exactly the kind of artifact a developer would produce in practice. Recent work such as RPG [4] follows a similar pipeline—produce documentation first, then generate code and tests—and TestExplora directly evaluates Stage 2 of this pipeline. Even if an LLM achieves strong performance on our benchmark, we acknowledge that real-world conditions may introduce noise from imperfect documentation. However, we believe evaluating under a clean-specification setting is a necessary first step: it isolates the LLM's test-generation capability from confounding factors, providing a meaningful and interpretable signal.
> > >
> > >
> > > $Ref$
> > >
> > >
> > > [1] https://aws.amazon.com/cn/documentation-overview/kiro/
> > >
> > > [2] https://github.blog/ai-and-ml/generative-ai/spec-driven-development-with-ai-get-started-with-a-new-open-source-toolkit/
> > >
> > > [3] https://claude.com/blog/using-claude-md-files
> > >
> > > [4] RPG: A Repository Planning Graph for Unified and Scalable Codebase Generation, ICLR26.

---

### Official Review · Reviewer_uuKR · 2026-03-13

**Soundness:** 3
**Presentation:** 3
**Significance:** 3
**Originality:** 3
**Overall Recommendation:** 3
**Confidence:** 3

**Summary:**

The paper introduces TestExplora, a repository-level benchmark designed to evaluate the capability of LLMs in proactive bug discovery. It evaluates if models can uncover latent defects using only implementation code and documentation-derived intent as an oracle. The benchmark contains 238 tasks from 482 repositories. Evaluation shows a significant performance gap of state-of-the-art LLMs.

**Compliance With Llm Reviewing Policy:**

Affirmed.

**Final Justification:**

The rebuttal did not address some of my main concerns.

**Key Questions For Authors:**

1. The authors claim that proactive bug discovery is 'largely unexplored' in current evaluations (Lines 043-075). However, the software engineering community has recently produced several benchmarks and frameworks evaluating LLMs and agents for proactive static bug detection and vulnerability discovery (e.g., LLift, LLM4PFA, KNighter, IRIS, and JitVul). These tools also analyze codebases to find defects before they manifest as failures. Could the authors justify their novelty claims to explicitly distinguish TestExplora from existing static analysis and fuzzing benchmarks?

2. The suite-level F2P metric inherently penalizes models that generate a higher volume of tests (e.g., Gemini-2.5-pro), as generating more tests mathematically increases the probability of a single false positive failing the head commit. Does this metric inadvertently incentivize models to generate minimal, superficial tests rather than thorough test suites?

3. In a truly proactive, repository-wide discovery scenario, a tester does not know where the latent bug resides. By explicitly providing the model with the exact 'Test entry inferences' (E) that lead to the bug (Section 3.2), isn't the benchmark implicitly leaking the location of the vulnerability? Could the authors discuss how narrowing the search space to these specific entry points impacts the claim of repository-scale proactive discovery?

**Limitations:**

The paper did not discuss the limitations. The paper should at least discuss the limitations of reproducibility relevant to GitHub actions, Doc generation etc.

**Strengths And Weaknesses:**

Strengths:
- Large scale up-to-date dataset. The dataset contains 2000+ tasks from 482 repositories.
- Insightful analysis. The paper identifies that "generating more tests" does not correlate with success and highlights
- Leakage prevention. The time-aware data collection framework is a proactive solution to the known problem of LLM training data contamination.

Weaknesses:
- Reliance on documentation generated by LLMs. The use of DocAgent introduces a potential bias where models might perform better because they are tested against intent generated by another LLM rather than human-written specifications.
- Complexity of reproducibility. Building the virtual environments requires an automated GitHub Actions Runner setup. This highly dynamic environment setup could affect future researchers attempting to reproduce the benchmark exactly as underlying package managers evolve.
- Traditional proactive bug discovery methods, e.g., fuzzing, symbolic/concolic execution and their synergy with LLMs/agents have shown great performance in bug detection, but are not discussed and compared.

---

> ### Author Rebuttal · Authors · 2026-03-30
>
> # Response to Reviewer uuKR
> Below, we address your specific concerns.
>
> ## W1: Potential Bias of DocAgent
> * **Empirical evidence contradicts model bias**:
> 	* **Stylistic bias is unlikely:** As the o3-mini-generated documentation does not give GPT-4o an advantage. If stylistic bias existed, we would expect models from the same family as the documentation generator to disproportionately benefit, but Claude-Sonnet-4.5 outperforms all GPT variants.
> 	* **Cross-model validation confirms robustness:** At the same time, we use DeepSeek-R1 to generate the documentation with the same method for 100 random examples for test generation:
>
> | Model| GPT-5-mini   | o4-mini | o3-mini   | Gemini-2.5-pro | DeepSeek-R1 | Claude-Sonnet-4.5  |
> |-|-|-|-|-|-|-|
> |o3-mini-Doc F2P|11|7|7|5|4| 17|
> |R1-Doc F2P|10|8|6|5|4|15|
>
> * **Standardizing the Intent Oracle to Mitigate Sample Variance**: Human-written documentation in real-world repositories exhibits variance in quality, format, and completeness. Relying on it would actually introduce bias across different samples, conflating a model's testing capability with its ability to parse poor or missing documentation. We use DocAgent to ensure a standardized intent oracle across samples.
> * **Aligning with TDD Principles to Isolate Bug Discovery**: TestExplora adopts a TDD paradigm, which inherently requires detailed and explicit specifications. Providing generated, comprehensive specifications to isolate codebase reasoning tasks is an established practice, similar to the approach adopted by recent repository-level works like RPG [1].
>
> ## W2: Complexity of Reproducibility
>
> We clarify that our GitHub Actions setup **is fully reproducible** through standardized runner environments. We will also release Docker images with pinned dependency versions and a single-command reproduction script. We have internally verified full reproducibility across different machines.
>
> ## W3: Traditional Proactive Bug Discovery Methods
>
> We agree these methods are powerful, but they are often unsuitable for TestExplora. Traditional methods like Pynguin are limited to single-module unit tests.  Additionally, our manual execution on first 12 TestExplora samples (due to non-automation) shows that **Pynguin fails to generate any F2P tests**.
>
> ## Q1: Related Works
>
> We will cite them and explicitly contrast TestExplora with them in the Related Work section. The key distinctions are twofold:
> * Regarding the output paradigm, existing frameworks primarily generate static warnings or perform binary vulnerability classification (JitVul). **In contrast, TestExplora requires models to autonomously synthesize complete, executable test suites.**
> * Second, regarding detection targets, related works primarily focus on predefined security vulnerabilities or memory errors (e.g., specific CWEs or Linux bugs), whereas **TestExplora focuses on diverse real-world projects**.
>
> ## Q2: F2P Metric
>
> TestExplora pairs the F2P metric with coverage metrics to ensure comprehensiveness detailed in Section 3.3. Furthermore, empirical results demonstrate that highly capable models (Claude-Sonnet-4.5 with 23.94 tests and SWEAgent+GPT-5-mini with 23.09 tests) can generate a large volume of tests while still achieving high F2P and coverage scores.
>
> ## Q3: Test Entry Inferences
>
> Providing the entry inferences does not lead to defect leakage, nor does it undermine the proactive discovery setting:
> * **Real-World Testing Scenario**: In practical settings, testing is anchored to specific features or public APIs rather than blindly searching. Without these explicit anchors, the search space becomes an infinite needle in a haystack.
>
> * **Tool Usage Analysis**: An analysis of SWEAgent's tool invocation distribution confirms that both models rely on exploratory tools rather than direct generation:
>
> | Tool | view | submit | pytest | python | grep -R |
> |-|-|-|-|-|-|
> | GPT-5-mini |56.0%|7.3%|2.4%|2.0%|2.0%|
> | o4-mini |48.4%|8.1% |<0.2%|1.2%|11.5%|
>
> * **Relation between Entry Inferences and Bug Location**: We analyze the call graph distance between the actual bug locations and the test function in our benchmark. Crucially, ~42% of the tests require the model to trigger bugs located strictly deeper (distance > 1):
>
>
> | Distance | 1    | 2   | 3   | 4   | 5   | 6+   |
> |-|-|-|-|-|-|-|
> | Num.     | 1828 | 725  | 346  | 173   |  44  |  50  |
>
> * **Leakage Ablation**: Additionally, we run a control experiment where we explicitly leak the fix-patch to SWEAgent+GPT-5-mini. Under this forced leakage, the F2P success rate surges from 17.27% to 73.94%, proving that our setting effectively suppresses defect information:
>
> | Model      | Method    | HP | F2P   | EC | CFG | Num.  |
> |-|-|-|-|-|-|-|
> | gpt-5-mini | SWEAgent  | 93.81     | 17.27 | 65.45       | 47.43        | 23.09 |
> |  | SWEAgent (patch leakage)  | 98.26     | 73.94 | 59.13       | 53.73        | 34.33 |
>
> $Ref$
>
> [1] RPG: A Repository Planning Graph for Unified and Scalable Codebase Generation, ICLR26.

---

> > ### Author Rebuttal · Reviewer_uuKR · 2026-04-02
> >
> > Thanks the authors for addressing the issues. I have the below remaining concerns:
> >
> > Q1: In traditional bug discovery/vulnerability detection, a test input is also required and generated, for example, fuzzing. It can be used or transformed to a PoC (proof of concept) that help developers to understand the bug. They can not only discover memory-related bugs, but also large amount of semantic level bugs, e.g., protocol violation, XSS injection vulnerability etc.
> > Q2: Such empirical evidence cannot well address the threat of hacking the metric. For example, a model can easily cheat by generating a lot of simple/similar tests.

---

> > > ### Author Response · Authors · 2026-04-02
> > >
> > > # Response to Reviewer uuKR's Follow-up
> > >
> > > We sincerely thank Reviewer uuKR for the continued engagement and thoughtful follow-up. Below is our clarification.
> > >
> > > ## Q1: Traditional Bug Discovery Methods
> > >
> > > * **Clarification on Pynguin's Methodology.**: We first clarify that Pynguin itself employs search-based test generation algorithms combined with dynamic analysis and mutation-based assertion generation, which fall under the category of traditional techniques the reviewer mentioned (fuzzing, symbolic/concolic execution).
> > >
> > > * **Detailed Results on Initial Samples**: We provide the full Pynguin execution results on the first 12 TestExplora samples below. Crucially, none of the 12 samples achieve a Fail-to-Pass transition. Specifically, the test results are identical on both the base (buggy) and head (fixed) commits:
> > >
> > > |Sample|Base Execution Result|Head Execution Result|
> > > |-|-|-|
> > > |davidaurelio/hashids-python (4)|15 passed, 2 xfailed in 0.12s|15 passed, 2 xfailed in 0.08s|
> > > |mjpost/sacrebleu (134)|1 passed, 6 xfailed in 0.20s|1 passed, 6 xfailed in 0.19s|
> > > |vi3k6i5/flashtext (29)|17 passed, 10 xfailed in 0.20s|17 passed, 10 xfailed in 0.16s|
> > > |google/pinject (47)|1 failed, 9 passed, 5 xfailed in 0.09s|1 failed, 9 passed, 5 xfailed in 0.08s|
> > > |python-babel/babel (1135)| 18 passed, 15 xfailed in 0.71s | 18 passed, 15 xfailed in 0.65s |
> > > |cookiecutter/cookiecutter (2096)|6 passed, 9 xfailed in 0.49s|6 passed, 9 xfailed in 0.48s|
> > > |msiemens/tinydb (279)|1 passed, 6 xfailed in 0.22s|1 passed, 6 xfailed in 0.22s|
> > > |PyCQA/isort (2264)|`Test Generation Failed`|` Test Generation Failed`|
> > > |konradhalas/dacite (4)|2 failed, 4 xfailed in 0.10s|2 failed, 4 xfailed in 0.07s|
> > > |hbldh/bleak (1744)|4 passed in 0.05s | 4 passed in 0.04s|
> > > |jupyterhub/the-littlest-jupyterhub (1013)|` Test Generation Failed`|` Test Generation Failed`|
> > > |keras-team/keras-preprocessing (328)|5 passed, 7 xfailed, 1 warning in 0.21s|5 passed, 7 xfailed, 1 warning in 0.19s|
> > >
> > > * **Extended Evaluation**: To provide a more robust comparison, we extended the evaluation to 100 randomly sampled TestExplora instances:
> > >
> > > |Metric|Runnable Rate|F2P|EC|
> > > |-|-|-|-|
> > > |Pynguin|72|0|42.23|
> > >
> > > Pynguin achieves 0% F2P, confirming that traditional search-based methods fundamentally cannot detect the category of bugs targeted by TestExplora.
> > >
> > > * **Further Clarification**: We acknowledge that traditional methods can effectively detect certain semantic-level bugs where correctness is defined by mechanically checkable properties. However, TestExplora's bugs involve intent-specification discrepancies. For example, a function documented to raise KeyError for absent keys but silently returning 0. In this scenario, the execution never crashes, so a fuzzer has no oracle to flag it. Generating many simple assertions also cannot help, since no quantity of shallow checks will flag a valid but semantically wrong output. Successful detection requires analyzing the documented specification, identifying the prescribed behavior, and synthesizing a targeted assertion (e.g., pytest.raises(KeyError)). This is precisely the intent-driven capability TestExplora evaluates, complementary to traditional oracle-checkable bug detection. We will add this discussion in Related Work.
> > >
> > >
> > >
> > > ## Q2: Evaluation Metric
> > >
> > > TestExplora's multi-metric design resists such hacking. Our metrics form a complementary system:
> > > * F2P requires tests to actually trigger the latent bug: simple/similar tests that crash at the API boundary or never reach the fault path score 0.
> > > * EC requires broad coverage of the entry interface: repetitive tests exercising the same paths yield low EC.
> > > * CFG requires reaching the actual modified code: this demands navigating deep dependency chains, which trivial tests cannot achieve.
> > >
> > > Critically, these metrics create a mutual tension that prevents trivial optimization of any single metric:
> > > * Reducing test volume to maximize F2P inevitably lowers EC, since fewer tests cannot comprehensively cover the entry interface.
> > > * Inflating test volume with simple/similar tests to boost EC will lower both F2P (suite-level penalty from incorrect tests) and CFG (shallow, repetitive tests fail to penetrate deep enough to reach the bug location).
> > >
> > > For example, Pynguin's superficial tests on `konradhalas/dacite` achieve only F2P=0 and EC=0.33, confirming our framework's resistance to hacking via simple/similar and superficial tests.
> > >
> > > ```
> > > # Test cases automatically generated by Pynguin (https://www.pynguin.eu).
> > > # Please check them before you use them.
> > > import pytest
> > > import tinydb.database as module_0
> > >
> > >
> > > @pytest.mark.xfail(strict=True)
> > > def test_case_0():
> > >     none_type_0 = None
> > >     storage_proxy_0 = module_0.StorageProxy(none_type_0, none_type_0)
> > >     storage_proxy_0.write(storage_proxy_0)
> > >
> > > ...
> > >
> > > @pytest.mark.xfail(strict=True)
> > > def test_case_4():
> > >     str_0 = "or9OVvnqy"
> > >     module_0.DataProxy(str_0, str_0)
> > >
> > >
> > > def test_case_5():
> > >     none_type_0 = None
> > >     module_0.StorageProxy(none_type_0, none_type_0)
> > > ```

---

### Decision · Program_Chairs · 2026-04-30

**Decision:**

Accept (regular)

**Comment:**

We thank the authors for their submission. Reviewers broadly agreed that the problem framing in this work is worthwhile, and appreciated the large dataset constructed from diverse sources. They also welcomed the analysis controlling for time to assess the risk of memorization.

There was significant discussion around the design decision to rely on documentation produced from the golden patch. This design decision helps ensure that the task is well-defined and solvable, but decreases its realism. In practical use, models are virtually never presented with this type of oracle information. In this case, however, the primary goal is to evaluate how well existing LLMs can diagnose a mismatch. Within that framing, the design is reasonable. Given that they largely perform very poorly at this task, it is also reasonable to track their potential on the harder, real-world task by including this type of privileged information in the input at this time. The authors are encouraged to make it clear that performance on this task will not translate exactly to real-world scenarios.

Separately, reviewers noted that the work misses references to some  related work from, among others, the test generation literature, and includes a number of methodological decisions that may introduce risks or that are interpreted as unreasonably strong safeguards. The authors are strongly encouraged to adjust the language and acknowledgements on these fronts.